# Complement Decay-Accelerating Factor is a modulator of influenza A virus lung immunopathology

**Nuno Brito Santos**[1☯], **Zoé Enderlin Vaz da Silva**[1☯], **Catarina Gomes**[2,3], **Celso A. Reis**[2,3,4,5], **Maria João Amorim**[1] *

**1** Cell Biology of Viral Infection Lab, Instituto Gulbenkian de Ciência, Oeiras, Portugal, **2** Instituto de Investigação e Inovação em Saúde (i3S), University of Porto, Porto, Portugal, **3** Institute of Molecular Pathology and Immunology of the University of Porto (IPATIMUP), Porto, Portugal, **4** Molecular Biology Department, Institute of Biomedical Sciences of Abel Salazar (ICBAS), University of Porto, Porto, Portugal, **5** Pathology Department, Faculty of Medicine, University of Porto, Porto, Portugal

☯ These authors contributed equally to this work.
* mjamorim@igc.gulbenkian.pt

**Data Availability Statement:** All data are available at https://doi.org/10.5281/zenodo.4773828.

**Funding:** This work was funded by Instituto Gulbenkian de Ciência (IGC), Fundação Calouste

## Abstract

Clearance of viral infections, such as SARS-CoV-2 and influenza A virus (IAV), must be fine-tuned to eliminate the pathogen without causing immunopathology. As such, an aggressive initial innate immune response favors the host in contrast to a detrimental prolonged inflammation. The complement pathway bridges innate and adaptive immune system and contributes to the response by directly clearing pathogens or infected cells, as well as recruiting proinflammatory immune cells and regulating inflammation. However, the impact of modulating complement activation in viral infections is still unclear. In this work, we targeted the complement decay-accelerating factor (DAF/CD55), a surface protein that protects cells from non-specific complement attack, and analyzed its role in IAV infections. We found that DAF modulates IAV infection *in vivo*, via an interplay with the antigenic viral proteins hemagglutinin (HA) and neuraminidase (NA), in a strain specific manner. Our results reveal that, contrary to what could be expected, DAF potentiates complement activation, increasing the recruitment of neutrophils, monocytes and T cells. We also show that viral NA acts on the heavily sialylated DAF and propose that the NA-dependent DAF removal of sialic acids exacerbates complement activation, leading to lung immunopathology. Remarkably, this mechanism has no impact on viral loads, but rather on the host resilience to infection, and may have direct implications in zoonotic influenza transmissions.

## Author summary

Exacerbated complement activation and immune deregulation are at the basis of several pathologies induced by respiratory viruses. Here, we report that complement decay-accelerating factor (DAF), which inhibits complement activation in healthy cells, increases disease severity upon influenza A virus (IAV) infection. Remarkably, DAF interaction with IAV proteins, hemagglutinin (HA) and neuraminidase (NA), resulted in excessive

Gulbenkian (FCG) and Fundação para a Ciência e a Tecnologia (FCT) (PTDC/IMI-MIC/1142/2012). NBS was funded by Graduate Programme Science for Development (PGCD) and FCG. ZEVS was funded by FCT (SFRH/BD/52179/2013). CG was funded by FCT (POCI-01-0145-FEDER-029780, PTDC/MED-QUI/29780/2017). CAR was funded by FCT (POCI-01-0145-FEDER-007274, UID/BIM/04293). MJA is funded by FCT (2020.02373.CEECIND). The funders had no role in study design, data collection and analysis, decision to publish, or preparation of the manuscript.

**Competing interests:** The authors have declared that no competing interests exist.

complement activation and recruitment of innate and adaptive immune cells, without affecting viral loads. Furthermore, we observed that viral NA directly cleaves DAF and promotes complement activation, providing a possible link between IAV-DAF interaction and pathology. Therefore, our results unveil a novel pathway that could modulate disease severity, which may help to understand the increased pathogenicity of zoonotic and pandemic IAV infections.

## Introduction

Host-pathogen interactions are very complex with both parts contributing to the progression and outcome of infections. In the case of viruses, pathogen- and damage-associated molecular patterns (PAMP and DAMP, respectively) are detected by pattern recognition receptors (PRR) alerting the host of their presence, and triggering the immune response to clear the infection [1,2]. It is generally accepted that for viral infections, an aggressive initial activation of innate immunity favors the host, whilst mechanisms that originate prolonged inflammation are associated with severe outcomes. This paradigm underpins for example the sex differences observed for coronavirus disease 19 (COVID-19), that results in lower death rate in women, despite similar incidence of infection in both sexes [3–5]. However, an excessive immune response activation might destabilize the equilibrium needed to eliminate the pathogen without causing tissue damage, and lead to immunopathology [6,7]. It is therefore important to determine the host factors and viral characteristics that result in an efficient immune response for clearing the pathogen without provoking immunopathology.

Influenza A virus (IAV) is the prevalent cause of seasonal flu, a relevant health problem as it kills up to 600,000 people worldwide yearly [8]. IAV replication occurs in the upper and lower respiratory tract, peaks normally 2 days after infection, and in most cases little virus shed can be detected after 6 days. For the majority of people, symptoms (fever, cough, acute viral nasopharyngitis, headache) clear after 7–10 days, with fatigue enduring for weeks, without serious outcomes [8–10]. In a proportion of people, however, severe complications occur, with the elderly, immunosuppressed, pregnant women, and people with associated comorbidities being at higher risk [11]. IAV can also provoke pandemic outbreaks, associated with zoonotic events, which lead to significant higher mortality than seasonal epidemics. The 1918 Spanish influenza, for example, caused up to 50 million deaths [12]. Complications may include hemorrhagic bronchitis, pneumonia (primary viral or secondary bacterial), and death [13–16]. They usually derive from an exacerbated immune response leading to tissue damage [17,18]. Identifying intrinsic risk factors that contribute to severe disease outcomes may minimize immunopathology in the lungs and uncover new therapeutic targets with decreased proneness to develop resistance.

Defects in type I interferon (IFN) response have been associated with the more severe cases of COVID-19 [19,20], suggesting that the initial steps in immune activation define disease outcome. However, there are other players involved in mounting immune responses, such as the complement system. The complement system has been extensively reviewed [21–23] and consists in a cascade of proteolytic interactions that lead to the direct killing of the pathogen or infected cell, as well as proinflammatory immune cell recruitment. Remarkably, C3, central player in the complement cascade, has been found within the mucus barrier [24], which elucidates complement role in early immune response upon pathogen infection in the airways. Disease severity and mortality have been associated with both lack or excess of complement activation in several viral infections such as Severe Acute Respiratory Syndrome Coronavirus

(SARS-CoV) [25,26], Middle Eastern Respiratory Syndrome Coronavirus (MERS-CoV) [27], SARS-CoV-2 [28–30], and IAV [31–33]. However, it is still unclear how fine-tuning complement activation may impact the development of disease severity. One strategy to tune complement activation in infection is to target its regulators. Complement decay-accelerating factor (DAF/CD55) is a membrane-bound regulator of complement activation (RCA) exposed at the surface of most cell types, including human and murine airways [34–36]. DAF promotes the decay of C3 convertases, thus protecting healthy cells from non-specific complement attack, and inhibiting the release of anaphylatoxins that would recruit and activate the immune response [37–39]. In humans, it has been reported that DAF deficiency leads to excess complement activation with systemic implications [40,41]. Furthermore, SNPs in DAF promoter region decreasing protein expression have been associated with higher risk of severe infections by pandemic and avian IAV strains [42,43].

In this work, we explore the role of DAF in activating complement and in modulating IAV infection via an interplay with the antigenic viral proteins hemagglutinin (HA) and neuraminidase (NA). We observed that DAF, contrary to what could be expected, potentiates complement activation in IAV infection. We also describe that viral NA acts on DAF, in a strain-specific manner, removing α2,6-linked sialic acids and propose that this may influence pathogenicity. Given that the recognition of different conformations of sialic acid by the influenza virus is a key driver in influenza intra- and interspecies transmission, our findings may have implications for zoonotic events. Our results also showed that DAF leads to increased complement activation, as well as immune cell recruitment, especially of neutrophils and monocytes, increasing lung immunopathology without altering viral loads. Our work reveals a novel mechanism of virulence in IAV infection.

## Results

### Decay-accelerating factor (DAF) aggravates IAV infection by increasing immunopathology

Immune response to viral infections such as IAV must be tightly regulated in order to clear the pathogen without causing immunopathology. The complement system is at the frontline of the immune response, recognizing pathogens, and activating and recruiting immune cells. The absence of a regulator of this system, such as DAF, is expected to increase complement activation, resulting in more efficient viral clearance and/or increased tissue damage. To assess the consequences of DAF depletion in the context of IAV infection, C57BL/6J (WT) and C57BL/6J $Daf^{-/-}$ ($Daf^{-/-}$) mice were challenged with two different H1N1 strains circulating in the human population: A/California/7/2009 (Cal) and A/England/195/2009 (Eng). For each viral strain, we measured bodyweight loss, as proxy for disease severity, and followed survival up to 11 days post infection (d.p.i.) (Fig 1). Surprisingly, we observed that upon infection with Cal, $Daf^{-/-}$ mice exhibited reduced bodyweight loss starting at 4 d.p.i., when compared to the WT, and maintained that difference throughout the experiment (Fig 1A). In addition, percentage of survival of $Daf^{-/-}$ mice when infected with Cal was higher than of WT mice (75% vs. 25%) (Fig 1B). Similarly, upon challenge with Eng, $Daf^{-/-}$ mice had increased survival (55.6 % vs. 33.3%), but lost more of their initial bodyweight when compared to the WT (Fig 1C and 1D). WT mice surviving to Eng infection had a milder bodyweight loss when compared to $Daf^{-/-}$ mice, thus explaining the reversion in trends later in infection, and the consequent discrepancy between bodyweight loss and survival. Taken together, our data indicate that DAF exerts a detrimental effect for the host during IAV infection.

We then extended the observation to two different well-characterized IAV strains: the mouse adapted virulent H1N1 A/Puerto Rico/8/1934 (PR8) and the less virulent H3N2 A/X-

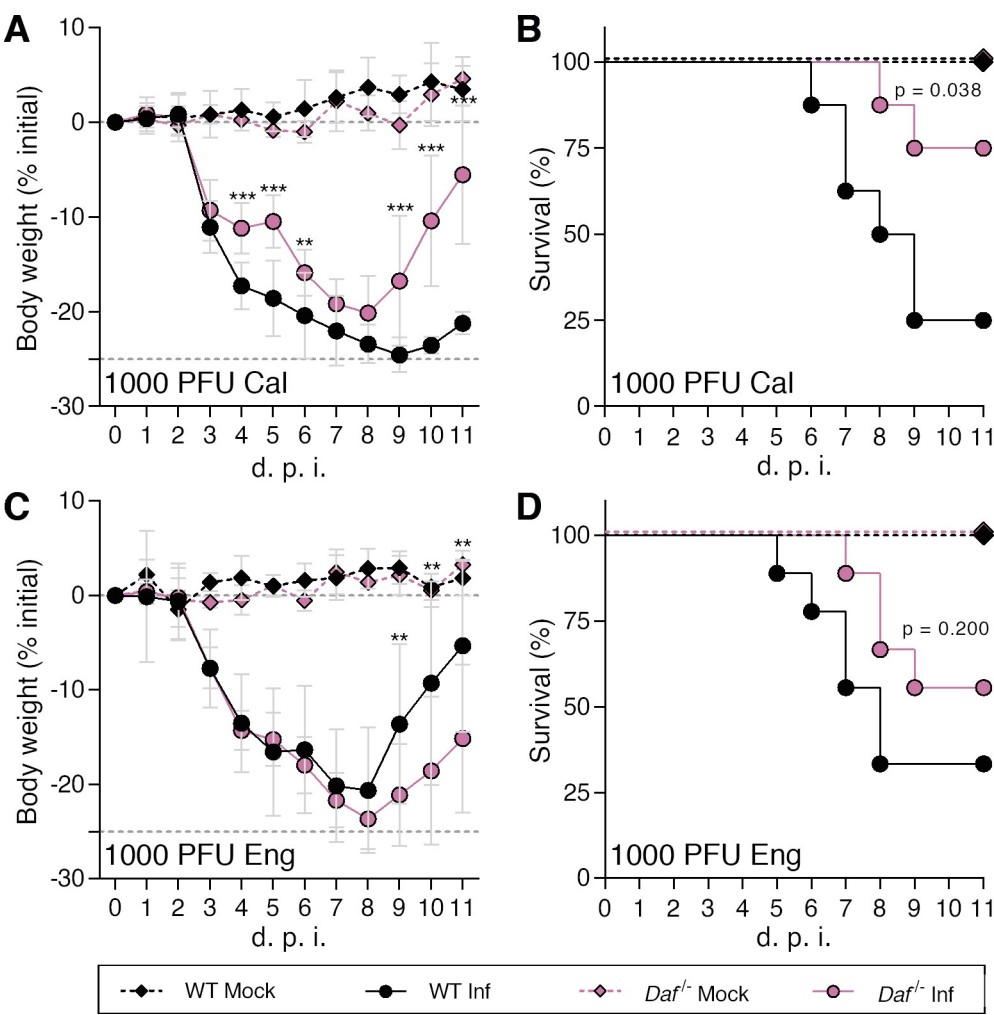

**Fig 1. Decay-accelerating factor (DAF) aggravates IAV infection *in vivo*. A, B**: Bodyweight loss (**A**) and mortality (**B**) of C57BL/6J WT or *Daf*^−/− mice infected with 1000 PFU of A/California/7/2009 (Cal) (Inf n = 8; mock n = 3 per group) **C, D**: Bodyweight loss (**C**) and mortality (**D**) of C57BL/6J WT or *Daf*^−/− mice infected with 1000 PFU of A/England/195/2009 (Eng) (Inf n = 9 per group; mock n = 5 and n = 4 for WT and *Daf*^−/− respectively). Results are expressed as mean±sd, statistical analysis detailed in materials and methods.

31 (X31). X31 is a reassortant strain of PR8 containing segments 4 and 6 from A/Hong Kong/1/68 (HK68) [44] and for clarity purposes, the X31 strain will be mentioned as PR8-HK4,6 throughout this work. WT and *Daf*^−/− mice were infected with sublethal and lethal doses of PR8 or PR8-HK4,6, and bodyweight loss and mortality assessed for 11 d.p.i.. Upon infection with PR8, *Daf*^−/− and WT mice presented similar bodyweight loss and all mice succumbed to the disease, *Daf*^−/− mice at 7 d.p.i., and WT mice at 8 d.p.i. (Fig 2A and 2B). Upon infection with PR8-HK4,6, as observed in infections with the circulating strains, *Daf*^−/− mice had a less severe disease and mortality when compared with their WT counterparts. These mice lost less of their initial bodyweight (-11.3% vs. -20.4%) and had reduced mortality than WT mice (50% vs. 100%) (Fig 2C and 2D). The discrepancy between PR8 and the other strains might be explained by the high virulence of this strain where 500 PFU of PR8 are a lethal dose, here quantified by the humane endpoint of infection of a loss of more than 25% of initial bodyweight. These results show that DAF worsens disease outcome in infection with mildly virulent

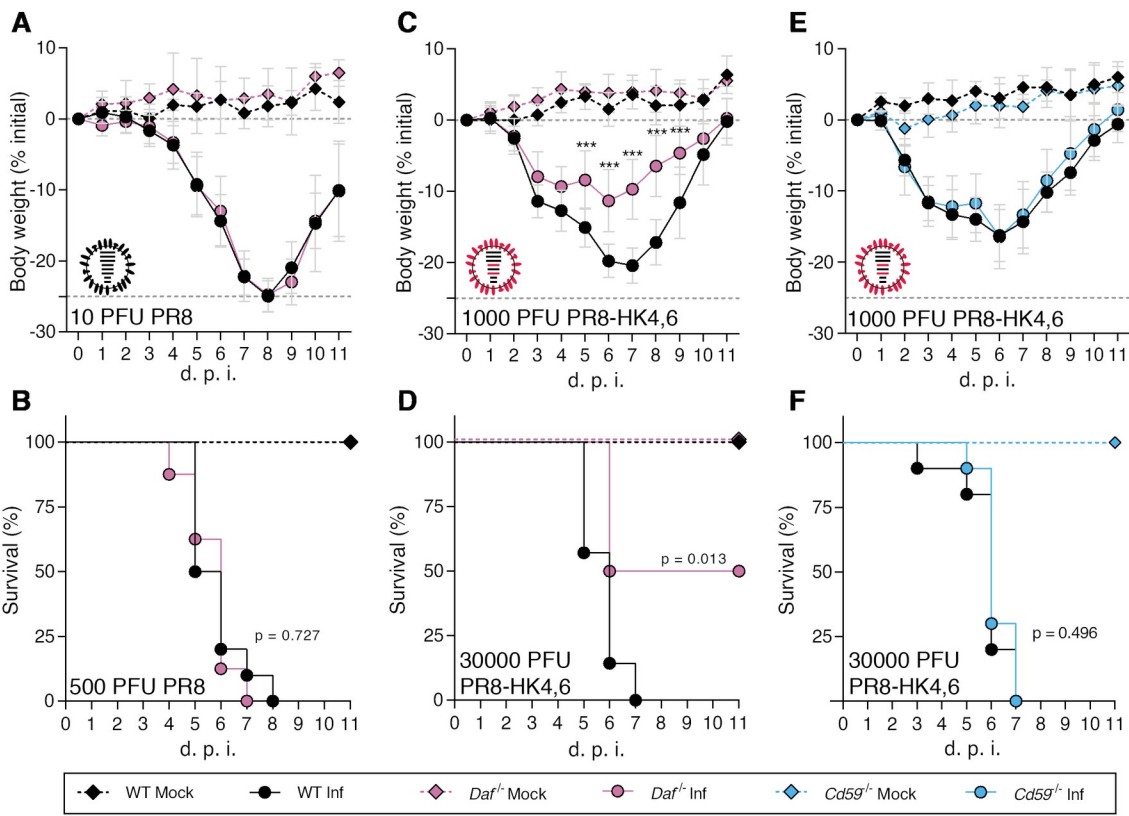

**Fig 2. *Daf<sup>-/-</sup>* mice are protected against PR8-HK4,6, but not PR8, and protection is specific of this RCA. A, B**: Bodyweight loss (**A**) and mortality (**B**) of C57BL/6J WT or *Daf<sup>-/-</sup>* mice infected with the indicated doses of A/Puerto Rico/8/1934 (PR8) (**A**: Inf n = 8 per group; mock n = 7 and n = 4 for WT and *Daf<sup>-/-</sup>* respectively; **B**: Inf n = 9 and n = 10, mock n = 8 and n = 4 for WT and *Daf<sup>-/-</sup>* respectively). **C, D**: Bodyweight loss (**C**) and mortality (**D**) of C57BL/6J WT or *Daf<sup>-/-</sup>* mice infected with the indicated doses of A/X-31 (PR8-HK4,6) (**C:** Inf n = 9 and n = 10, mock n = 8 and n = 4 for WT and *Daf<sup>-/-</sup>* respectively; **D:** Inf n = 7 and n = 8, mock n = 3 and n = 2 for WT and *Daf<sup>-/-</sup>* respectively). **E, F**: Bodyweight loss (**E**) and mortality (**F**) of C57BL/6J WT or *Cd59<sup>-/-</sup>* mice infected with the indicated doses of PR8-HK4,6 (**E:** Inf n = 10 and n = 11 for WT and *Cd59<sup>-/-</sup>* respectively, mock = 7 per group; **F:** Inf n = 10 per group, mock n = 4 and n = 7 for WT and *Cd59<sup>-/-</sup>* respectively). Results are expressed as mean±sd, statistical analysis detailed in materials and methods.

IAV strains, both circulating in the human population and lab-adapted, but not with more virulent IAV strain.

As DAF is a complement regulator, the results indicate a role for complement in modulating disease outcome. To dissect between the role of the complement pathway and of this particular molecule, C57BL/6J mice depleted of CD59 (*Cd59<sup>-/-</sup>*), another regulator of complement activation (RCA), were infected with PR8-HK4,6, and bodyweight loss and mortality assessed for 11 d.p.i.. Interestingly, there was no difference in bodyweight loss and mortality between *Cd59<sup>-/-</sup>* mice and their WT counterparts, with all mice succumbing to the disease at 7 d.p.i. (Fig 2E and 2F). This indicates that the protection observed in *Daf<sup>-/-</sup>* mice is mediated by the absence of this particular RCA, and not due to a general complement deregulation.

Taken together, our results suggest a role for DAF in disease outcome. To further dissect the mechanisms behind such role, we focused on infections with PR8-HK4,6 as it is a well-described laboratorial model, with a virulence resembling circulating strains.

Protection conferred by DAF depletion could be explained by a decrease in viral burden or by preventing immunopathology [6]. To distinguish between these two hypotheses, we started by assessing lung viral loads of WT and *Daf<sup>-/-</sup>* mice infected with a sublethal dose of

PR8-HK4,6. Samples were collected at day 2, 3, 6, 7 and 9 p.i. to distinguish between early viral replication and clearance. At days, 2, 3 and 7 p.i., lung viral titers were identical in WT and $Daf^{-/-}$ mice ($2.2 \pm 1.1 \times 10^6$ PFU/g vs. $1.4 \pm 0.7 \times 10^6$ PFU/g at 2 d.p.i., $3.8 \pm 2.8 \times 10^6$ PFU/g vs. $3.1 \pm 2.7 \times 10^6$ PFU/g at 3 d.p.i., $3.2 \pm 2.7 \times 10^5$ PFU/g vs. $3.3 \pm 4.3 \times 10^5$ PFU/g at 6 d.p.i., $2.3 \pm 3.9 \times 10^4$ PFU/g vs. $1.5 \pm 3.9 \times 10^4$ at 7 d.p.i.). At 9 d.p.i. no viruses were detected (Fig 3A). Thus, the amelioration of disease outcome is not associated with reduced viral replication or faster clearance. We then interrogated if the difference observed between WT and $Daf^{-/-}$ mice could be explained by a spatial difference in lung tissue infection, as was previously described for milder disease progression [45]. To detect infected cells in specific parts of the lung tissue, we performed immunohistochemistry (IHC) staining of viral nucleoprotein (NP) in mice lung sections at 3 d.p.i., time corresponding to higher viral loads. A blind qualitative observation elucidated that in both WT and $Daf^{-/-}$ mice, infection foci were mainly restricted to alveoli (Fig 3B), and quantification of infected bronchioli per lung section did not display relevant dissimilarities ($27.8 \pm 12.7$ in WT and $28.4 \pm 9.8$ in $Daf^{-/-}$) (Fig 3C).

Lastly, to assess if protection of $Daf^{-/-}$ mice was linked to a decrease in lung damage and immunopathology, a comprehensive and blind histological analysis of lung tissue was performed at day 3, 6 and 9 p.i. (Figs 3D and S1 and S1 Table). At an early time point, 3 d.p.i., $Daf^{-/-}$ mice had a histological score of $4.0 \pm 1.3$, whereas WT mice had a score of $5.4 \pm 2.0$. At 6 d. p.i., which corresponds to the time point of greater difference in weight loss between the two strains of mice, the difference in histological score became significant, with $Daf^{-/-}$ mice having a score of $10.8 \pm 2.2$, and WT of $12.8 \pm 2.6$. Later in infection, at 9 d.p.i., mice presented a similar histological score of $8.0 \pm 3$ in WT mice and $8.1 \pm 3.5$ in $Daf^{-/-}$. These results indicate that reduced disease severity in $Daf^{-/-}$ mice infected with PR8-HK4,6 can be due to less tissue damage. As tissue damage is linked to inflammation and infiltration of immune cells, we analyzed alveolar, interstitial and perivascular/peribronchiolar inflammations over the course of infection (S1 Table). The sum of these parameters was used as a semi-quantitative measure of cell infiltration in the lung parenchyma and analyzed at days 3, 6 and 9 p.i. (Fig 3E). However, the levels of cell infiltration in the lung parenchyma were comparable in WT and $Daf^{-/-}$ mice over the course of infection.

Taken together, our data shows that DAF does not impact viral replication, clearance nor spatial distribution in the lungs, but point to a new role for DAF as an immunopathology instigator.

## DAF-induced immunopathology relies on elevated complement activation, immune cell recruitment and antiviral response

We have shown that $Daf^{-/-}$ mice suffer less severe disease than WT mice upon IAV infection by decreasing tissue damage. Next, we aimed at dissecting the mechanism. DAF being an RCA, we first focused on determining the role of the complement pathway. For that purpose, C57BL/6J $C3^{-/-}$ ($C3^{-/-}$) and C57BL/6J $C3^{-/-}$ / $Daf^{-/-}$ ($C3^{-/-}$ / $Daf^{-/-}$) mice were infected with 1000 PFU of PR8-HK4,6 and bodyweight loss monitored over the course of infection. As expected, $C3^{-/-}$ mice lost significantly more bodyweight than the WT [46,47], losing up to 19.9% of the initial bodyweight, when WT mice lost only 17.8%. $C3^{-/-}$ / $Daf^{-/-}$ mice, however, had a bodyweight loss comparable with $C3^{-/-}$ mice, losing up to 18.1% of the initial bodyweight (Fig 4A). These results show that the protection of $Daf^{-/-}$ mice is C3-dependent, and thus complement mediated.

DAF regulates complement activation by accelerating the decay of C3 convertases, reducing the levels of C3a. Hence, we proceeded by analyzing the levels of C3a in the bronchoalveolar lavages (BALs) of PR8-HK4,6 infected WT or $Daf^{-/-}$ mice. C3a levels were analyzed at 2, 3, 6

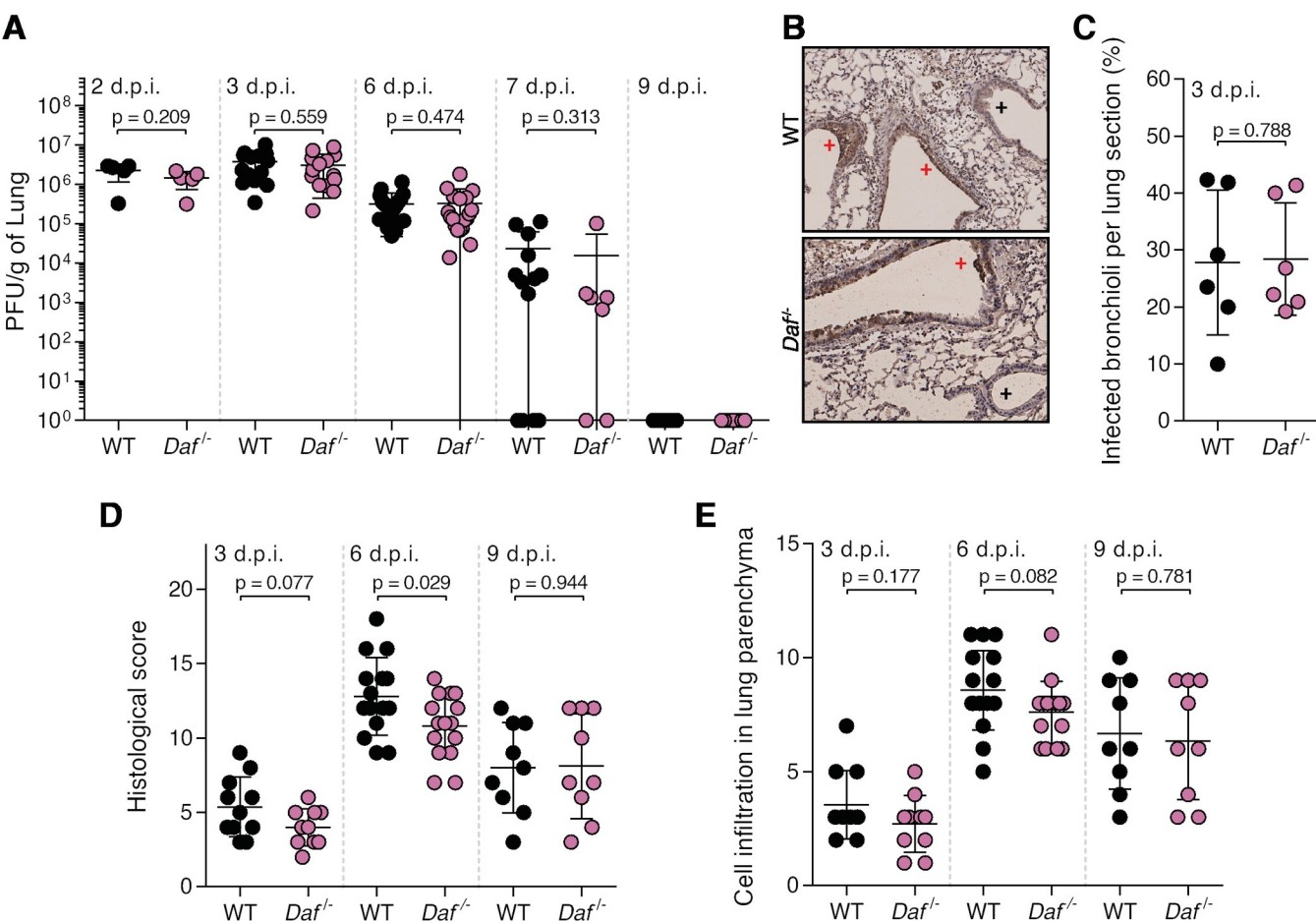

**Fig 3. DAF does not affect viral replication, clearance, or tissue penetration, but is an immunopathology instigator. A:** Lung viral titers of C57BL/6J WT or *Daf*<sup>-/-</sup> mice infected with 1000 PFU of A/X-31 (PR8-HK4,6). Samples collected at 2 d.p.i. (n = 5 per group); 3 d.p.i. (n = 14 per group), 6 d.p.i. (n = 20 and n = 19 for WT and *Daf*<sup>-/-</sup> respectively), 7 d.p.i. (n = 13 and n = 7 for WT and *Daf*<sup>-/-</sup> respectively) and 9 d.p.i. (n = 9 per group). **B**: Immunohistochemistry detection of IAV nucleoprotein (NP) in WT or *Daf*<sup>-/-</sup> mice 3 d.p.i. with 1000 PFU of PR8-HK4,6 (+ healthy; + infected). **C**: Quantification of infected bronchioli (n = 6 per group). **D, E**: Histological score (**D**) and cell infiltration in lung parenchyma (**E**) of C57BL/6J WT or *Daf*<sup>-/-</sup> mice infected with 1000 PFU of PR8-HK4,6 was assessed blindly. Evaluated parameters detailed in S1 Table. Samples collected at 3 d.p.i. (n = 11 and n = 10 for WT and *Daf*<sup>-/-</sup> respectively), 6 d.p.i. (n = 15 per group) and 9 d.p.i. (n = 9 per group). Results are expressed as mean±sd. Statistical analysis detailed in materials and methods.

and 7 d.p.i. to account for early activation and persistence of said activation (Fig 4B). At day 2 p.i. levels of C3a in BALs of *Daf*<sup>-/-</sup> and WT mice were similar (116.2±95.2 ng/mL vs. 214.4 ±111.1 ng/mL). Quite surprisingly, at day 3 p.i. *Daf*<sup>-/-</sup> mice had lower levels of C3a indicating less complement activation in these mice when compared to their WT counterparts, with *Daf*<sup>-/-</sup> having 63.6±50.4 ng/mL and WT 399.4±595.0 ng/mL. This difference was maintained at days 6 and 7 p.i. with C3a levels of 476.7±167.8 ng/mL vs. 1425±899.5 ng/mL and 291.8±120.3 ng/mL vs. 1013±30.07 ng/mL in *Daf*<sup>-/-</sup> and WT mice respectively. These results show that IAV infection induced lower complement activation in *Daf*<sup>-/-</sup> mice than in WT mice, thus indicating that complement activation may play a role in increased tissue damage of WT mice. Taken together, these results highlight the equilibrium needed to clear the disease without causing damage and the important role of complement in both these processes.

The complement pathway is a cascade of reactions that will release cytokines for recruitment and activation of the immune system, and culminating in the formation of a cytolytic pore (C5b-9). Our results showed that deleting *Cd59*, inhibitor of C5b-9, does not impact

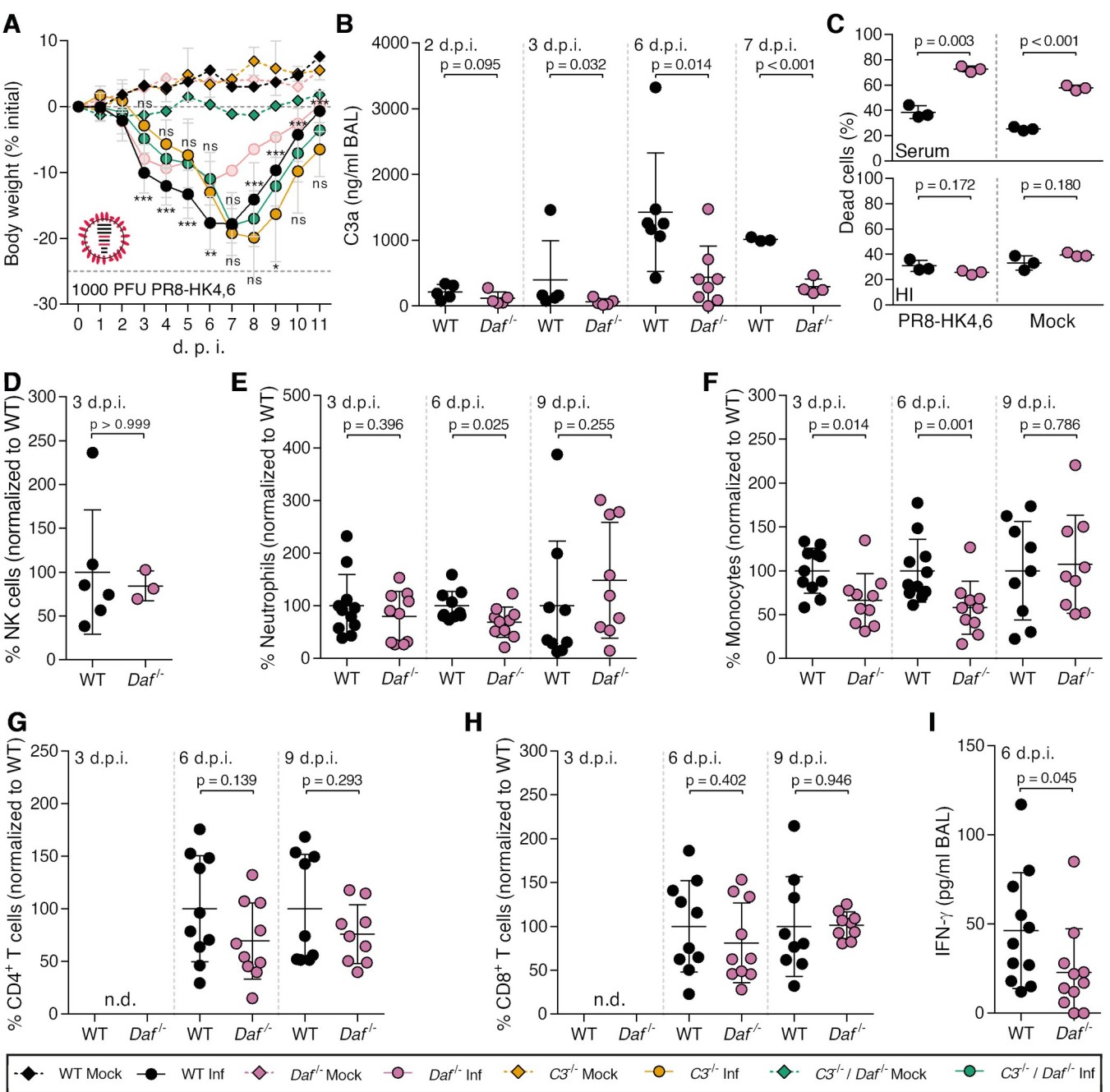

**Fig 4.** ***Daf*<sup>-/-</sup> mice have reduced complement activation and recruitment of innate immune cells. A:** Bodyweight loss of C57BL/6J WT, *Daf*<sup>-/-</sup>, *C3*<sup>-/-</sup> or *C3*<sup>-/-</sup> / *Daf*<sup>-/-</sup> mice infected with 1000 PFU of A/X-31 (PR8-HK4,6) (Inf n = 10, n = 10, n = 6 and n = 10; mock n = 9, n = 4 n = 5 and n = 7 for WT, *Daf*<sup>-/-</sup> *C3*<sup>-/-</sup> and *C3*<sup>-/-</sup> / *Daf*<sup>-/-</sup> respectively). Results are expressed as mean±sd. **B:** C3a levels in BALs of C57BL/6J WT or *Daf*<sup>-/-</sup> mice 2 (n = 5 per group), 3 (n = 5 per group), 6 (WT n = 7; *Daf*<sup>-/-</sup> n = 8) and 7 (WT n = 3; *Daf*<sup>-/-</sup> n = 4) d.p.i. with 1000 PFU of PR8-HK4,6. Results are expressed as mean±sd. **C:** Cell death of primary lung cells derived from WT or *Daf*<sup>-/-</sup> mice infected or mock-infected with PR8-HK4,6 and treated with serum. Results are expressed as mean±sd from 3 replicates from 2 independent experiments. **D, E, F, G, H:** Analysis of NK cells (**D**), neutrophils (**E**) monocytes (**F**) CD4<sup>+</sup> T cells (**G**) and CD8<sup>+</sup> T (**H**) cells levels in BALs of WT or *Daf*<sup>-/-</sup> mice infected with 1000 PFU PR8-HK4,6. Samples were collected at 3 d.p.i. (**D**, n = 6 and n = 3; **E—H**, n = 11 and n = 10 for WT and *Daf*<sup>-/-</sup> respectively), 6 d.p.i. (n = 10 per group) and 9 d.p.i. (n = 9 per group). Samples with non detectable cell levels are noted as n.d.. **K:** IFN-γ levels in BALs of WT or *Daf*<sup>-/-</sup> mice, 6 d.p.i. with 1000 PFU PR8-HK4,6 (n = 10 per group). Results are expressed as mean±sd from at least 2 independent experiments. A representative experiment with absolute values can be found in S3A–S3D Fig. Statistical analysis detailed in materials and methods.

disease outcome in the context of IAV infection (Fig 2E and 2F), suggesting that the protection observed in $Daf^{-/-}$ mice does not rely on complement-dependent cytotoxicity (CDC). To confirm this hypothesis, WT and $Daf^{-/-}$ murine primary lung cells were infected with PR8-HK4,6, treated with serum collected from naïve WT mice, and cell viability assessed as a measurement of CDC (Fig 4C). $Daf^{-/-}$ derived lung cells were more prone to CDC than WT-derived ones, both at steady state (57.7±2.1% vs. 25.4±1.5%) and upon PR8-HK4,6 infection (72.6±2.3% vs. 38.5±5.1%). This effect is specific of complement attack, as heat-inactivated serum did not increase cell death (Fig 4C), and confirms that $Daf^{-/-}$ mice protection is not dependent on complement cytolytic attack.

Given that $Daf^{-/-}$ mice have lower complement activation but that protection does not depend on CDC, it should rely on the release of anaphylatoxins leading to an alteration of immune cell recruitment and/or activation. Although differences in cell infiltration in lung parenchyma between WT and $Daf^{-/-}$ mice were not significant, there was a clear trend of increased levels in WT mice (Fig 3E). To complement our results and fully characterize the recruitment of immune cells to the lung we assessed the levels of specific immune cell types in BALs of WT and $Daf^{-/-}$ mice were infected with 1000 PFU of PR8-HK4,6. Analyses were carried at 3, 6 and 9 d.p.i. in order to uncouple the first rapid response from a more mature later one. At 3 d.p.i. we observed that $Daf^{-/-}$ mice had similar numbers of natural killer (NK) cells and neutrophils recruited to the lungs, when compared to WT mice (84.4±16.8% vs. 100 ±71.0% NK cells; 79.7±47.4% vs. 100±59.3% neutrophils), but lower numbers of monocytes (66.3±30.3% vs. 100±25.6%) (Fig 4D–4F). At 6 d.p.i., $Daf^{-/-}$ mice maintained the lower number of monocytes when compared to WT mice (58.1±30.3% vs. 100±35.8%), and also had reduced levels of neutrophils (69.1±28.8% vs. 100±27.5%) (Fig 4E and 4F). Levels of NK cells were not analyzed at this time point, nor in following analysis, as depletion of NK cells in PR8-HK4,6 infected WT mice did not alter disease outcome (S2A and S2B Fig). At a later time point in infection, 9 d.p.i., WT and $Daf^{-/-}$ mice had similar levels of neutrophils and monocytes (100.0 ±123.1 vs 148.6±109.9 neutrophils and 100.0±56.1 vs 107±55.8 monocytes) (Fig 4E and 4F). Therefore, $Daf^{-/-}$ mice recruit less neutrophils and monocytes to the lungs early in infection, which could explain the reduced histological score observed in these mice (Fig 3D). Additionally, we analyzed the recruitment of adaptive immune cells, namely CD4$^+$ and CD8$^+$ T cells that have been shown to play an important role in IAV infection [48]. As expected, at day 3 p.i. we were not able to detect T cells in the BALs of WT and $Daf^{-/-}$ mice infected with PR8-HK4,6 [49,50]. Interestingly, at both 6 d.p.i. and 9 d.p.i. there was no difference in recruitment of both CD4$^+$ and CD8$^+$ T cells (Fig 4G and 4H), indicating that the protection observed in $Daf^{-/-}$ mice is likely dependent on lower immunopathology mediated by the innate immune response.

Cytokines are also key players in the recruitment and activation of the immune system. IFN-γ, in particular, is an essential player in viral responses, and, like all members of the immune system, can cause tissue damage. Indeed, it has recently been shown that IFN-γ, which is produced upon IAV infection, is detrimental to the host by suppressing the protective effect of group II innate lymphoid cells (ILC2) [51]. Therefore, levels of IFN-γ were measured in BALs of PR8-HK4,6-infected WT and $Daf^{-/-}$ mice at 6 d.p.i.. $Daf^{-/-}$ mice had significantly lower levels of IFN-γ than WT (22.9±24.3 pg/mL vs. 44.4±32.5 pg/mL) (Fig 4I), which is in accordance with the reduced immunopathology and tissue damage in this context.

Given this result, we thought to further explore the contribution of DAF in modulating inflammatory cues upon viral infection. For that purpose, we assessed a panel of antiviral cytokines at days 3, 6 and 9 p.i. with PR8-HK4,6 in BALs of WT and $Daf^{-/-}$ mice (Fig 5). We observe that the content of type I IFN (IFN-α and IFN-β) was not significantly decreased in $Daf^{-/-}$ mice at all tested time points (Fig 5A and 5B). As type I IFN play an important role in

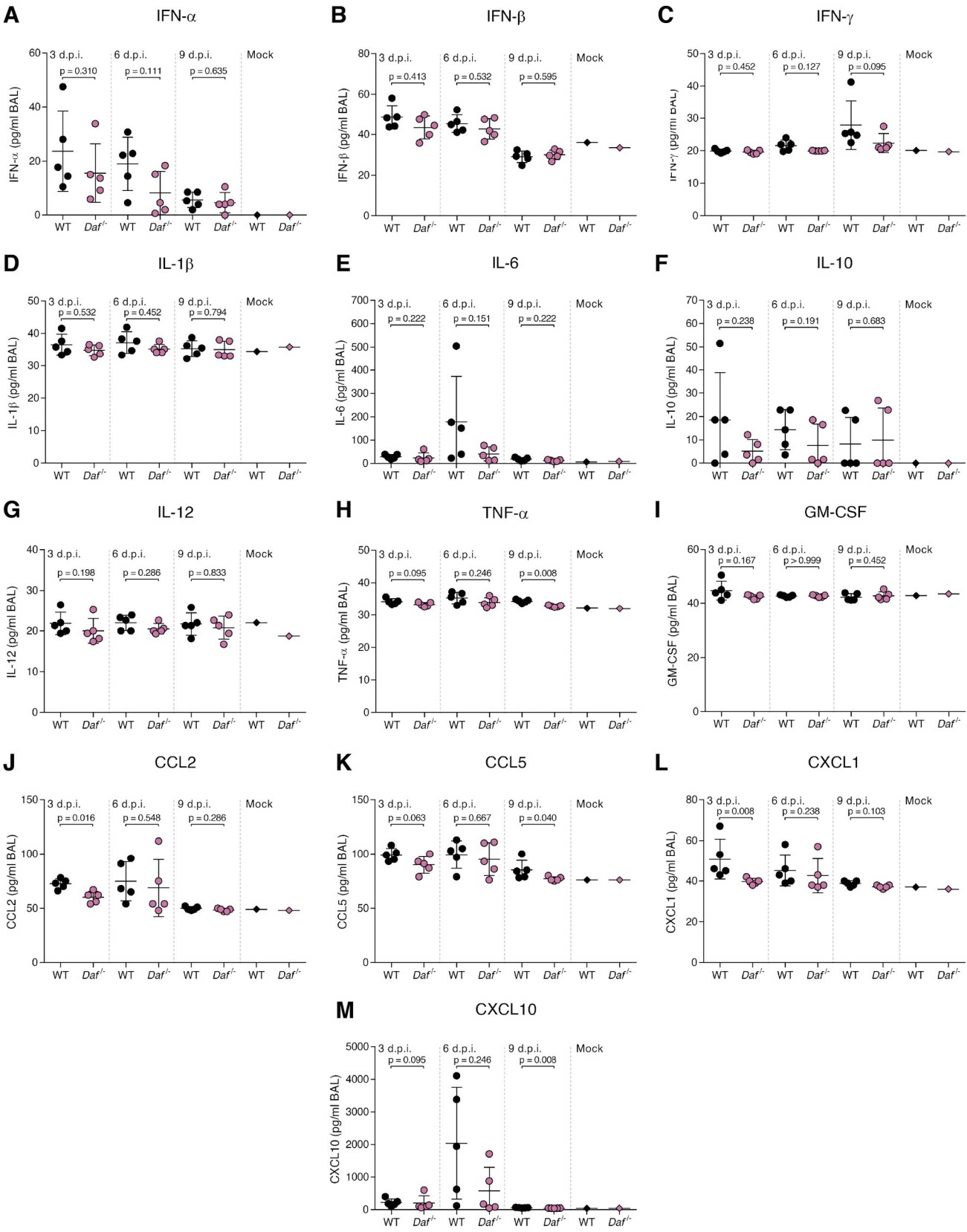

**Fig 5. *Daf*<sup>-/-</sup> mice present overall lower antiviral cytokine response.** Antiviral cytokine levels were measured in BALs of C57BL/6J WT or *Daf*<sup>-/-</sup> mice infected with 1000 PFU of PR8-HK4,6 at 3, 6 and 9 d.p.i. (n = 5 per group). **A:** interferon (IFN)-α, **B:** IFN-β, **C:** IFN-γ, **D:** interleukin (IL)-1β, **E:** IL-6, **F:** IL-10, **G:** IL-12, **H:** tumor necrosis factor (TNF)-α, **I:** granulocyte-macrophage colony-stimulating factor (GM-CSF), **J:** C-C motif chemokine ligand (CCL) 2, **K:** CCL5, **L:** C-X-C motif chemokine ligand (CXCL) 1, **M:** CXCL10. Results are expressed as mean±sd from at least 2 independent experiments. Statistical analysis detailed in materials and methods.

controlling viral replication [52], this result is in agreement with the similar viral loads observed in WT and *Daf*<sup>-/-</sup> mice over the course of PR8-HK4,6 infection (Fig 3A). Using this system, we observed a trend of lower IFN-γ levels in *Daf*<sup>-/-</sup> mice at 9 d.p.i. compared to that observed in WT, that however, did not reach statistical significance due to the variations within the 5 animals analyzed per condition (Fig 5C).

Interleukins (IL) also contribute to excessive inflammation and immunopathology [53]. Interestingly, IL-1β, IL-6, IL-10 and IL-12 were not significantly altered upon DAF depletion over the course of infection (Fig 5D and 5G). However, and albeit not significant, we observed an increase in IL-6 content in the BALs of WT mice at 6 d.p.i., that is absent in *Daf*<sup>-/-</sup> (179.0 ±193.8 pg/mL vs. 41.2±29.7 pg/mL) (Fig 5E). Remarkably, this corresponds to the time point when WT mice present higher lung tissue damage and C3a levels when compared to *Daf*<sup>-/-</sup> mice, as well as higher numbers of neutrophils and macrophages recruited to the lungs (Figs 3D, 4B, 4E and 4F). This increase of IL-6 in BALs of WT infected mice could thus explain the increased immune cell recruitment leading to tissue damage and immunopathology.

The potent proinflammatory cytokine tumor necrosis factor (TNF)-α showed an overall trend of being decreased in *Daf*<sup>-/-</sup> mice, being significantly lower at 9 d.p.i. (34.3±0.5 pg/mL vs. 32.7±0.3 pg/mL) (Fig 5H), suggesting that DAF contributes to maintain longer inflammation.

Although no relevant alterations were observed for the granulocyte-macrophage colony stimulating factor (GM-CSF) (Fig 5I), *Daf*<sup>-/-</sup> mice had significantly reduced levels of C-C motif chemokine ligand (CCL)2 (or monocyte chemoattractant protein-1, MCP-1) (Fig 5J) at 3 d.p.i.. Moreover, *Daf*<sup>-/-</sup> mice did not present the abrupt increase in C-X-C motif chemokine ligand (CXCL)10 observed in WT mice at 6 d.p.i. and had significantly lower levels of this chemokine at 9 d.p.i. (49.7±3.8 pg/mL vs. 63.7±8.7 pg/mL) (Fig 5M). CCL2 and CXCL10 being monocyte chemoattractants [54–57], the lower levels of these chemokines in the lungs would explain the reduced numbers of monocytes in PR8-HK4,6 infected *Daf*<sup>-/-</sup> mice when compared to the WT. Interestingly, at 3 d.p.i. *Daf*<sup>-/-</sup> mice also had significantly lower levels of CXCL1. Although we observe reduced neutrophils in *Daf*<sup>-/-</sup> mice at 6 d.p.i. (Fig 4E), it is tempting to speculate that this initial reduction in CXCL1 will result in lower neutrophil recruitment later in infection. Similarly, at 9 d.p.i., *Daf*<sup>-/-</sup> mice had lower levels of CCL5 (or regulated on activation, normal T cell expressed and secreted, RANTES) than WT mice (77.0±2.0 pg/mL vs. 85.2±8.8 pg/mL) (Fig 5K). Although we did not observe significant alterations in CD4<sup>+</sup> or CD8<sup>+</sup> T cell recruitment (Fig 4G and 4H), it is tempting to speculate that reduced levels of CCL5 would lead to a faster resolution of the inflammation and faster reduction of T cell levels at a later time point.

Taken together, these results indicate that lower complement activation leads to a reduced antiviral response and recruitment of innate immune cells, such as neutrophils and monocytes. This will allow a reduction in tissue damage, ameliorating disease outcome. Interestingly, and counter-intuitively, the decrease in complement activation is a consequence of the absence of a major complement regulator, DAF.

## DAF-induced immunopathology depends on viral HA and NA

We observed that lack of DAF protected mice from infection with PR8-HK4,6, but not with PR8 (Fig 2A–2D). These strains differ only in hemagglutinin (HA) and neuraminidase (NA)

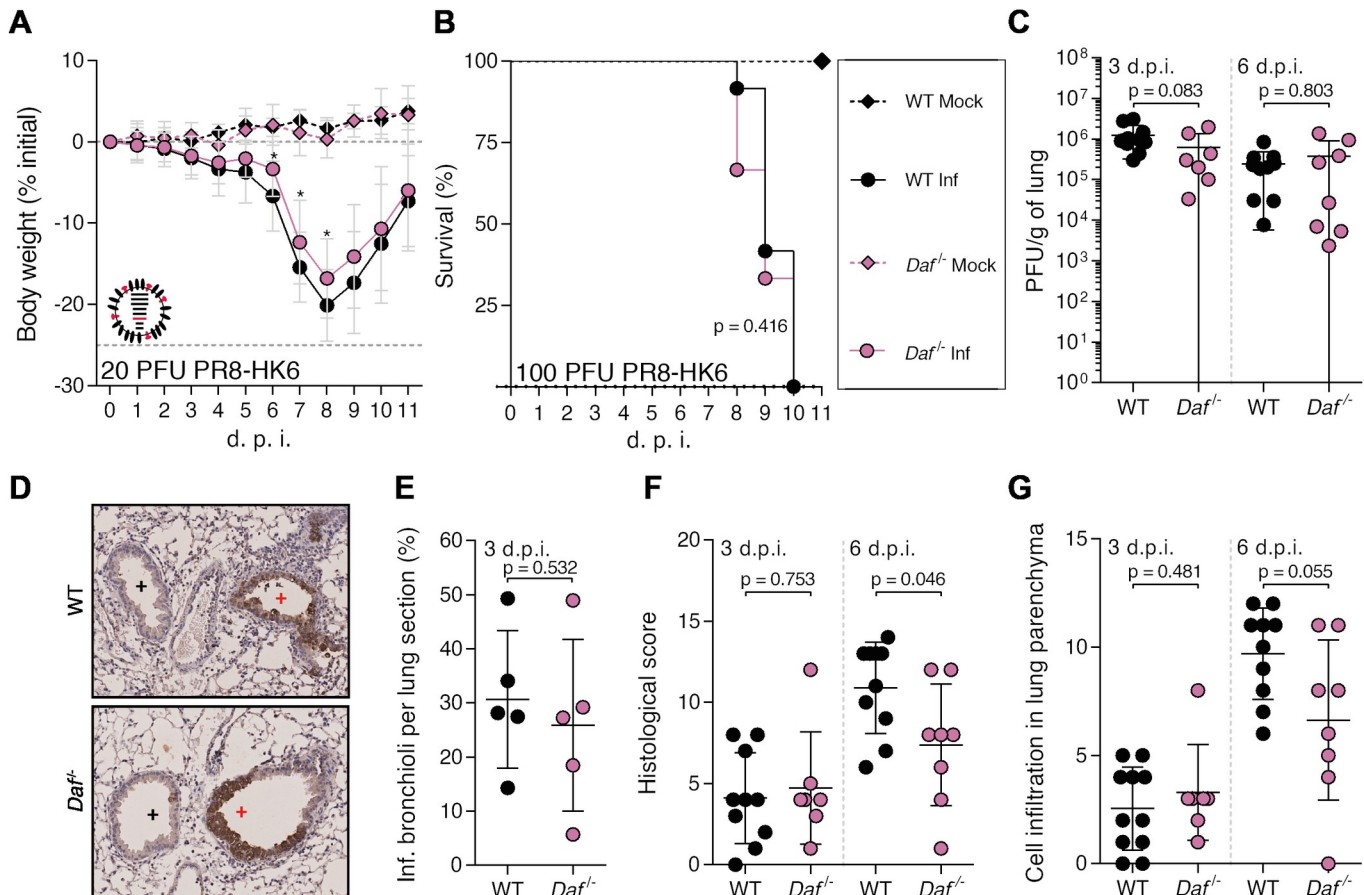

**Fig 6. DAF interaction with HA worsens disease outcome, without increasing immunopathology. A, B**: Bodyweight loss (**A**) and mortality (**B**) of C57BL/6J WT or *Daf*<sup>/-</sup> mice infected with the indicated doses of A/Puerto Rico/8/1934 with segment 6 from A/Hong Kong/1/68 (PR8-HK6). (**A:** Inf n = 16 and n = 18, mock n = 6 and n = 7 for WT and *Daf*<sup>/-</sup> respectively; **B:** Inf n = 12 and n = 9, mock n = 4 and n = 2 for WT and *Daf*<sup>/-</sup> respectively). Results are expressed as mean±sd. **C**: Lung viral titers of C57BL/6J WT or *Daf*<sup>/-</sup> mice infected with 20 PFU of PR8-HK6. Samples collected at 3 and 6 d.p.i. (n = 10 and n = 7 for WT and *Daf*<sup>/-</sup> respectively). **D**: Immunohistochemistry detection of IAV nucleoprotein (NP) in WT or *Daf*<sup>/-</sup> 3 d.p.i. with 20 PFU of PR8-HK6 (+ healthy; + infected). **E**: Quantification of infected bronchioli (n = 5 per group). **F, G**: Histological score and cell infiltration of lung parenchyma of C57BL/6J WT or *Daf*<sup>/-</sup> mice infected with 20 PFU of PR8-HK6. Samples collected at 3 d.p.i. (n = 10 and n = 7 for WT and *Daf*<sup>/-</sup> respectively) and 6 d.p.i. (n = 10 and n = 8 for WT and *Daf*<sup>/-</sup> respectively). Results are expressed as mean±sd. Statistical analysis detailed in materials and methods.

[44]. To investigate the individual role of these proteins in the resilience to infection, we constructed chimeric viruses in PR8 background containing either HA (PR8-HK4) or NA (PR8-HK6) from HK68. It is important to note that analyses were performed in comparison with PR8-HK4,6 infections and not PR8. Therefore, it is the removal of HK6 in PR8-HK4 that will allow investigating the contributions of different NAs, and the removal of HK4 in PR8-HK6 that will enable assessment of the contributions of HAs. Furthermore, given the variability in virulence between viral strains, mice were infected with different PFUs of each virus as we aimed to reach similar disease progression, mortality and weight loss of WT mice, to be able to compare the impact of DAF depletion in these experiments.

On a first step, *Daf*<sup>/-</sup> and WT mice were infected with PR8-HK6, hence highlighting the role of HA. Infection with a sublethal dose of PR8-HK6 resulted in a modest amelioration of bodyweight loss in *Daf*<sup>/-</sup> mice, reaching -16.8% of the initial bodyweight, when compared to WT mice that lost up to 20.1% of the initial bodyweight (Fig 6A). When infected with lethal doses of this strain, both *Daf*<sup>/-</sup> and WT mice had a mortality of 100% (Fig 6B). *In vitro* and *ex*

*vivo* experiments had shown that this strain had increased replication levels when compared to PR8, PR8-HK4,6 or PR8-HK4 (S4A–S4E Fig). Therefore, we hypothesized that the increased mortality of $Daf^{-/-}$ mice, when compared to infection with the other strains, could be due to increased viral titers. Interestingly, analysis of lung viral loads showed no difference between $Daf^{-/-}$ and WT mice both at 3 and 6 d.p.i. (Fig 6C) and titers were not higher than those observed for the infection with PR8-HK4,6 (Fig 3A). Therefore, HA-DAF interaction modulates virulence, without impacting in viral replication or clearance *in vivo*.

As HA is involved in adhesion of viral particles to host cells, we asked if differences in HA would impact tissue penetration. As observed in PR8-HK4,6 infected mice, IHC of NP and quantification of infected bronchioli showed no difference in infection levels and patterns between $Daf^{-/-}$ and WT mice (Fig 6D and 6E), indicating that HA-DAF interaction has no role in this context. Additionally, analysis of tissue damage showed that histological scores between $Daf^{-/-}$ and WT mice were similar at 3 d.p.i. (4.7±3.5 vs. 4.1±2.8), but significantly reduced in $Daf^{-/-}$ mice when compared to WT at 6 d.p.i. (7.4±3.7 vs. 10.9±2.8) (Figs 6F and S1 and S1 Table). The high levels of lung damage showed in WT mice are supported by the analysis of cell infiltrates in lung parenchyma. Indeed, we observe no differences between WT and $Daf^{-/-}$ mice early in infection, but at 6 d.p.i., WT mice had increased cell infiltration when compared to $Daf^{-/-}$ (9.7±2.1vs 6.6±3.7, respectively) (Fig 6G). Therefore, as observed for PR8-HK4,6 infection, PR8-HK6 infected WT mice had higher levels of cell infiltrates in lung parenchyma and tissue damage when compared to their $Daf^{-/-}$ counterparts. These results show that HA-DAF interaction contributes to disease severity and worse disease outcome observed in WT mice, but does not impact lung tissue damage and hence does not completely explain the protective effect of DAF absence.

To better understand the role of HA-DAF interaction in disease outcome, we analyzed complement and immune cell recruitment in the lungs of PR8-HK6 infected mice. Interestingly $C3^{-/-}$ and $C3^{-/-}$ / $Daf^{-/-}$ mice had similar bodyweight loss when infected with PR8-HK6 (Fig 7A), and although the levels of C3a were identical between WT and $Daf^{-/-}$ mice early in infection (84.6±40.0 ng/mL vs 68.8±36.3 ng/mL), $Daf^{-/-}$ mice had reduced C3 than their WT counterparts later in infection (178.4±36.8 ng/mL vs. 405.8±99.2 ng/mL) (Fig 7B). These observations correspond to what was seen in PR8-HK4,6 infection and indicate that different HA-DAF interactions do not elicit different complement responses.

Analysis of lung immune cell recruitment in PR8-HK6 infected mice showed that at 3 d.p.i. levels of neutrophils and monocytes were identical between $Daf^{-/-}$ and WT mice (Fig 7C and 7D). At 6 d.p.i, however, $Daf^{-/-}$ mice had lower numbers of neutrophils and monocytes when compared to their WT counterparts (58.6±21.3% vs. 100±36.41% neutrophils; 61.4±19.6% vs. 100±41.45% monocytes) (Fig 7C and 7D) showing that a change in HA does not alter the innate immune cell recruitment observed in PR8-HK4,6. Of note, the levels of CD4+ and CD8+ T cells were decreased in PR8-HK6 infected $Daf^{-/-}$ mice when compared to their WT counterparts (57.5±26.2% vs. 100±35.9% CD4+ T cells and 49.3±36.7% vs. 100±50.5% CD8+ T cells) (Fig 7E and 7F), contrarily to what was seen in PR8-HK4,6 infection (Fig 4G and 4H) and showing that HA-DAF interaction may be implicated in modulating the adaptive immune response. This hypothesis requires further confirmation through the analysis of a set of markers in BAL over the course of infection.

Taken together, our data is consistent with HA-DAF interaction controlling disease severity, without impacting complement or innate immune responses leading to immunopathology. It does, however, impact the recruitment of T cells. The decreased activation of the adaptive immune response, together with the higher virulence of this strain may exceed the beneficial effect of reduced tissue damage and explain the similar mortality in $Daf^{-/-}$ and WT mice.

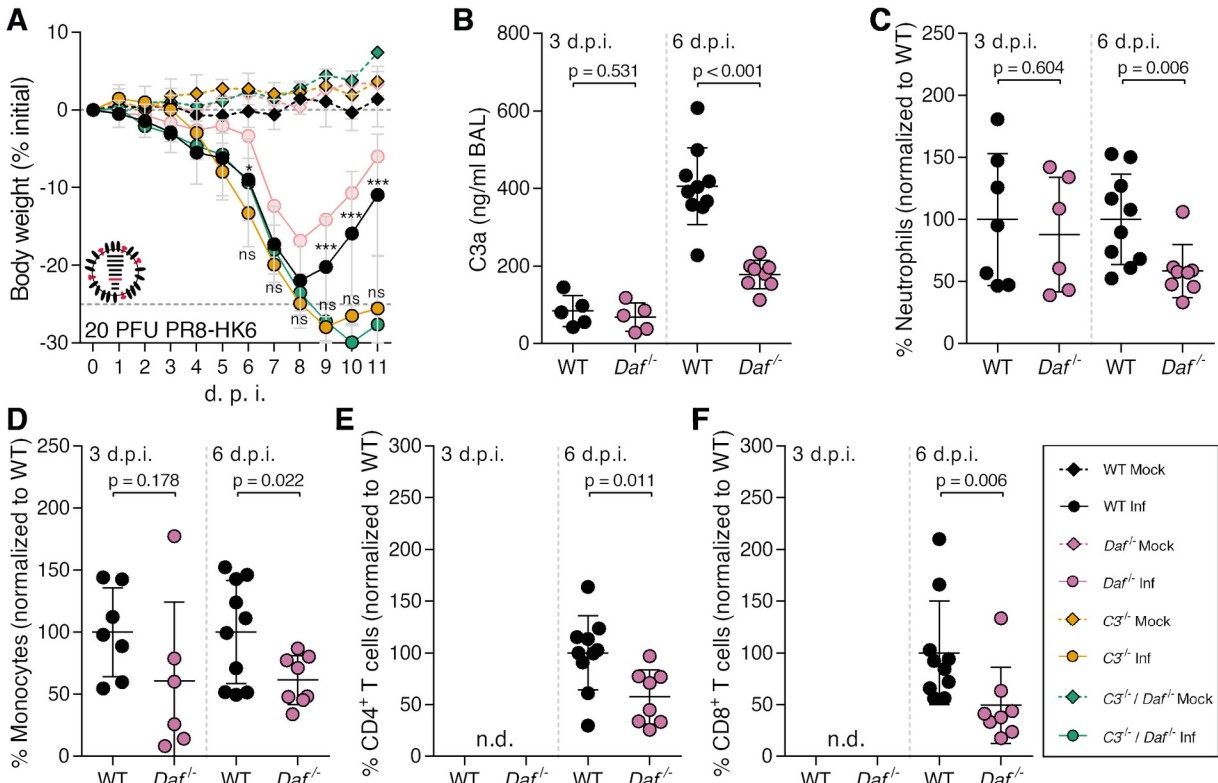

**Fig 7. *Daf⁻/⁻* mice have reduced complement activation and T cell recruitment upon PR8-HK6 infection. A**: Bodyweight loss of C57BL/6J WT, *Daf⁻/⁻*, *C3⁻/⁻* or *C3⁻/⁻ / Daf⁻/⁻* mice infected with 20 PFU of A/Puerto Rico/8/1934 with segment 6 from A/Hong Kong/1/68 (PR8-HK6) (Inf n = 26, n = 18, n = 10 and n = 4, mock n = 10, n = 7 n = 3 and n = 1, for WT, *C3⁻/⁻*, *C3⁻/⁻ / Daf⁻/⁻* and *Daf⁻/⁻* respectively). Results are expressed as mean±sd. **B**: C3a levels in BALs of C57BL/6J WT or *Daf⁻/⁻* mice at 3 d.p.i. (n = 5 per group) and 6 d.p.i. (WT n = 10 and *Daf⁻/⁻* n = 8) with 20 PFU of PR8-HK6. Results are expressed as mean±sd. **C, D, E, F**: Analysis of neutrophils (**C**), monocytes (**D**), CD4⁺ T cells (**E**) and CD8⁺ T cells (**F**) levels in BALs of WT and *Daf⁻/⁻* mice at 3 and 6 d.p.i. (n = 7 and n = 6 for WT and *Daf⁻/⁻* mice respectively) with 20 PFU PR8-HK6. Samples with non detectable cell levels are noted as n.d.. Results are expressed as mean±sd from two independent experiments. A representative experiment with absolute values can be found in S3E–S3H Fig. Statistical analysis detailed in materials and methods.

As HA-DAF interaction did not impact complement nor innate immune responses, we proceeded with analysis of NA-DAF interactions. Following the principle stated above, analyses were done in comparison with PR8-HK4,6 and not PR8 and thus the removal of HK6 from PR8-HK4,6 allowed assessing the role of different NAs. Therefore, to understand the contribution of NA in the protection conferred by DAF depletion, *Daf⁻/⁻* and WT mice were infected with sublethal and lethal doses of PR8-HK4. Upon infection with this strain, *Daf⁻/⁻* mice showed a reduced bodyweight loss when compared to WT mice (17.7% vs. 21.8% maximum bodyweight loss) (Fig 8A). The detrimental effect of DAF was more evident when mice were challenged with lethal doses of this strain. Indeed, 87.5% of WT mice succumbed to infection with 250 PFU of PR8-HK4, whereas all of *Daf⁻/⁻* mice survived (Fig 8B). As these results correspond to what was observed with PR8-HK4,6, NA-DAF interaction does not directly impact disease severity.

Similarly, lung viral loads were identical in *Daf⁻/⁻* and WT mice infected with PR8-HK4 both at 3 (2.2±1.9 x 10⁶ PFU/g vs. 4.1±5.5 x 10⁶ PFU/g) and 6 d.p.i. (8.8±9.9 x 10⁴ PFU/g vs. 6.1±4.1 x 10⁴ PFU/g) (Fig 8C). Also, PR8-HK4 infection foci were mainly restricted to the alveoli with no difference at the level of infected bronchioli in *Daf⁻/⁻* and WT mice lung sections (25.8±8.3% vs. 24.2±13.6%) (Fig 8D and 8E). These results show that NA-DAF interaction

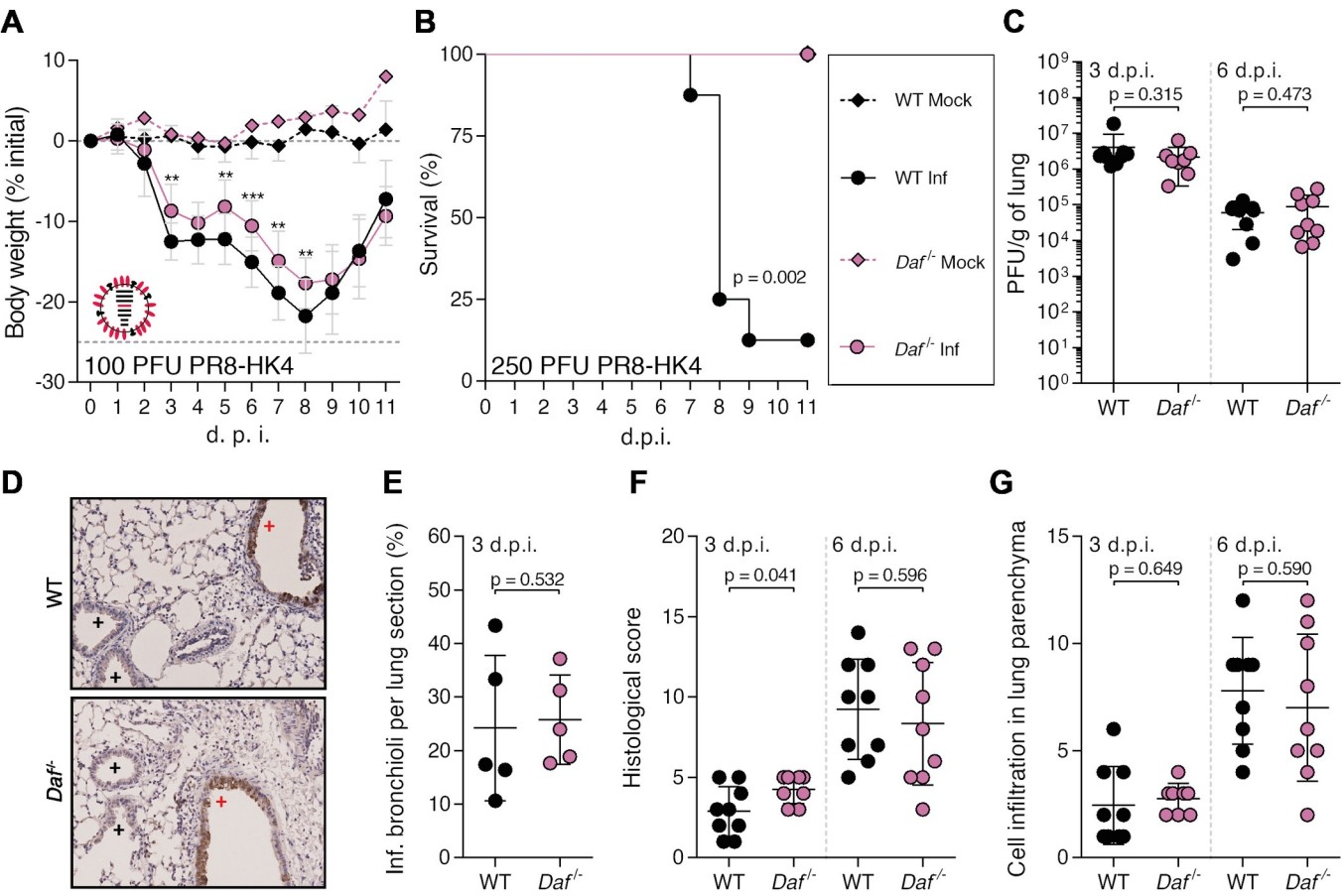

**Fig 8. DAF interaction with NA modulates immunopathology. A, B**: Bodyweight loss (**A**) and mortality (**B**) of C57BL/6J WT or *Daf*$^{-/-}$ mice infected with the indicated doses of A/Puerto Rico/8/1934 with segment 4 from A/Hong Kong/1/68 (PR8-HK4). (**A**: Inf n = 14 and n = 10, mock n = 4 and n = 1 for WT and *Daf*$^{-/-}$ respectively; **B**: Inf n = 8 per group, mock n = 4 and n = 1 for WT and *Daf*$^{-/-}$ respectively). Results are expressed as mean±sd**. C**: Lung viral titers of C57BL/6J WT or *Daf*$^{-/-}$ mice infected with 100 PFU of PR8-HK4. Samples collected at 3 d.p.i. (n = 9 and n = 8 for WT and *Daf*$^{-/-}$ respectively) and 6 d.p.i. (n = 9 per group). Results are expressed as mean±sd. **D**: Immunohistochemistry detection of IAV nucleoprotein (NP) in WT or *Daf*$^{-/-}$ 3 d.p.i. with 100 PFU of PR8-HK4 (+ healthy; + infected). **E**: Quantification of infected bronchioli (n = 5 per group). **F, G**: Histological score and cell infiltration in lung parenchyma of C57BL/6J WT or *Daf*$^{-/-}$ mice infected with 100 PFU of PR8-HK4. Samples collected at 3 d.p.i. (**F**, Inf n = 16 and n = 18, mock n = 6 and n = 7 for WT and *Daf*$^{-/-}$ respectively) and 6 d.p.i. (**G**, Inf n = 13 and n = 9, mock n = 4 and n = 2 for WT and *Daf*$^{-/-}$ respectively). Results are expressed as mean±sd. Statistical analysis detailed in materials and methods.

does not impact viral replication, clearance or tissue penetration. Interestingly, further analysis of PR8-HK4 infected lungs showed that the lungs of *Daf*$^{-/-}$ mice were more damaged at 3 d.p.i. with a histological score of 4.3±0.9, when compared to lungs from WT mice that had a score of 2.9±1.5. At 6 d.p.i. this difference was no longer present, *Daf*$^{-/-}$ lungs having a score of 8.3±3.8, and WT of 9.2±3.1 (Figs 8F and S1 and S1 Table). However, the levels of cell infiltrates in the lung parenchyma were similar in WT and *Daf*$^{-/-}$ mice both at 3 and 6 d.p.i. (Fig 8G). Thus, the increased lung damage observed in *Daf*$^{-/-}$ mice is not due to cell infiltration into the lung tissue. We showed that PR8-HK4 infected *Daf*$^{-/-}$ mice had more lung tissue damage at an early time point in infection (Fig 8F), when compared to WT mice, and oppositely to what was observed in PR8-HK4,6 infection (Fig 3D). NA-DAF interaction would then control lung immunopathology in this context, but with no real consequence in disease outcome, as *Daf*$^{-/-}$ still had reduced bodyweight loss and mortality when compared to the WT.

To better understand the mechanism behind this observation, we started by assessing the role of complement. *C3*$^{-/-}$ / *Daf*$^{-/-}$ and *C3*$^{-/-}$ mice had a similar bodyweight loss upon PR8-HK4

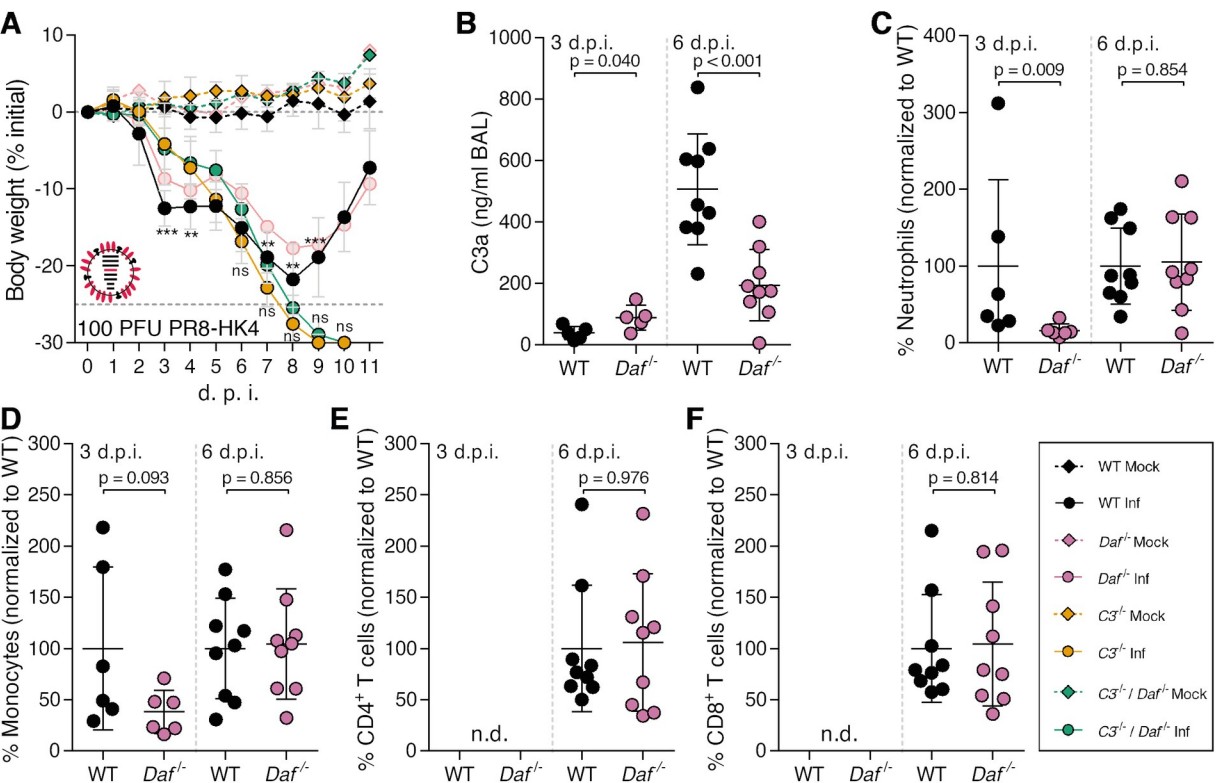

**Fig 9.** *Daf*$^{-/-}$ **mice present lower complement activation and neutrophil recruitment at 3 d.p.i. upon PR8-HK4 infection. A**: Bodyweight loss of C57BL/6J WT, *Daf*$^{-/-}$, *C3*$^{-/-}$ or *C3*$^{-/-}$ / *Daf*$^{-/-}$ mice infected with 100 PFU of A/Puerto Rico/8/1934 with segment 4 from A/Hong Kong/1/68 (PR8-HK4) (Inf n = 14, n = 10, n = 10, and n = 3, mock n = 4, n = 1, n = 3 and n = 1 for WT, *Daf*$^{-/-}$ *C3*$^{-/-}$, *C3*$^{-/-}$ / *Daf*$^{-/-}$ and *Daf*$^{-/-}$ respectively). Results are expressed as mean±sd. **B**: C3a levels in BALs of C57BL/6J WT or *Daf*$^{-/-}$ mice at 3 d.p.i. (n = 5 per group) 6 d.p.i. (n = 9 per group) with 100 PFU of PR8-HK4. Results are expressed as mean±sd. **C, D, E, F**: Analysis of neutrophils (**C**), monocytes (**D**), CD4$^+$ T cells (**E**) and CD8$^+$ T cells (**F**) levels in BALs of WT or *Daf*$^{-/-}$ mice, 3 d.p.i (n = 6 per group) and 6 d.p.i. (n = 9 per group) with 100 PFU PR8-HK4. Samples with non detectable cell levels are noted as n.d.. Results are expressed as mean±sd from two independent experiments. A representative experiment with absolute values can be found in S3I–S3L Fig. Statistical analysis detailed in materials and methods.

infection (Fig 9A). Surprisingly, C3a levels in BALs of PR8-HK4 *Daf*$^{-/-}$ mice were higher that their WT counterparts at 3 d.p.i. (39.4±21.5 ng/mL vs. 88.8±40.3 ng/mL), but, at 6 d.p.i. this trend was reversed and *Daf*$^{-/-}$ mice had lower C3a levels than WT (194.4±115.6 ng/mL vs. 506.4±180.2 ng/mL) (Fig 9B). These results confirm that, similarly to what was observed in PR8-HK4,6 and PR8-HK6 infections, the protection of *Daf*$^{-/-}$ mice upon PR8-HK4 infection is complement mediated. However, in this case, protection could be mediated by a higher complement response early in infection that will be rapidly controlled and decreased at later time points. This particular regulation of complement has been shown to be optimal for disease recovery in human viral infections [27–30]. To see if the different regulation of complement would affect immune cell recruitment, we proceeded with analysis of immune cell levels in the lungs at 3 and 6 d.p.i.. At 3 d.p.i., *Daf*$^{-/-}$ mice had reduced levels of neutrophils but not monocytes when compared to their WT counterparts (16.2±8.6% vs. 100±112.1% neutrophils; 38.1 ±21.1% vs. 100±79.5% monocytes) (Fig 9C and 9D). Then, at 6 d.p.i., *Daf*$^{-/-}$ and WT mice had comparable levels of both neutrophils and monocytes, and CD4$^+$ and CD8$^+$ T cells (Fig 9C–9F) (105.0±62.9% vs. 100±49.2% neutrophils; 104.5±54.0% vs. 100±49.1% monocytes; 106 ±67.0% vs. 100±61.7% CD4$^+$ T cells; 104.3±60.6% vs. 100±52.6% CD8$^+$ T cells). These results do not correspond to what was observed in infections with PR8-HK4,6, where the main differences between *Daf*$^{-/-}$ and WT mice resided in reduced numbers of monocytes at 3 d.p.i., and

reduced numbers of both neutrophils and monocytes at 6 d.p.i. (Fig 4E and 4F). We can therefore conclude that different NA elicit different innate immune responses, and that NA-DAF interaction is responsible for the recruitment of innate immune cells.

In summary, $Daf^{-/-}$ mice are protected from PR8-HK4 infection with decreased complement levels and reduced neutrophil recruitment but increased immunopathology early in infection. At later time points we did not observe differences between WT and $Daf^{-/-}$ mice regarding both lung tissue damage and immune cell recruitment. The reduction in neutrophil recruitment reflects what was observed in PR8-HK4,6 infection, albeit at an earlier time point. One might then suggest that NA-DAF interaction is important in regulation of neutrophil recruitment, and that these cells play an important role in modulating disease outcome. Taken together, our results demonstrate that both HA and NA play a role in disease severity, and that the cumulative effect of both HA- and NA-DAF interactions results in the mechanism worsening the outcome observed upon Cal, Eng and PR8-HK4,6 infections.

## Influenza A virus NA cleaves DAF through its sialidase activity

NA is a widely studied sialidase with described roles in mucus penetration, cell egress and recently even in viral entry [58]. Remarkably, NA has also been reported to cleave sialic acid residues from exogenous proteins inside the cell [59]. As DAF is a highly sialylated protein, we hypothesized that the interaction between DAF and NA resided in the ability of NA to cleave DAF's sialic acid content. Sialic acids that reside on cell surface glycoproteins and glycolipids are the receptors for IAV, recognized by HA for viral entry and cleaved by NA for viral exit [60]. In order to assess cleavage of DAF's sialic acid content, we infected a human alveolar cell line (A549) with Cal, Eng, PR8 and PR8-HK4,6, and analyzed DAF content by western blot. We observed that in infected cells the band marked by the anti-DAF antibody was at a lower molecular weight (MW) than in mock-infected cells (Fig 10A). This difference in MW is of nearly 18 kDa, which corresponds to DAF sialic acid content [61] and suggests that infection leads to loss of said content. Quantification of this cleavage confirmed that it is dependent on infection and progressive over time. Interestingly, the extent of DAF cleavage is not identical in cells infected with different IAV strains, PR8 infected cells presenting the most drastic effect (Fig 10B).

Protein glycosylation type and levels may greatly vary between organisms [62]. As previous results were obtained using human cell lines, we wanted to confirm that infection with the tested strains would remove the sialic acid content of murine DAF. For that purpose, we collected mouse embryonic fibroblasts (MEF) from WT mice and infected them with the laboratory adapted strains PR8 and PR8-HK4,6. Similarly to what was shown in a human cell line infection, murine DAF in infected MEFs suffered a drop in MW, when compared to non-infected cells (Fig 10C). Moreover, the differences in cleavage efficacy between PR8 and PR8-HK4,6 were maintained (Fig 10D), showing that IAV is able to process murine DAF and giving an insight to what may be triggering complement activation *in vivo*.

To show that NA mediates processing of DAF and discard the involvement of other viral proteins, we transfected HEK293T cells with eight different plasmids, each encoding a different PR8 genomic segment. As expected, cleavage only occurred when cells were transfected with segment 6, which encodes for NA, showing that NA is the only viral protein responsible for the reduction in DAF MW (Fig 10E).

As NA is a transmembrane protein with potential to cleave sialic acids at the cell surface, but also in the cytoplasm while en route to the plasma membrane, we questioned where DAF cleavage was taking place. For that, PR8 infected A549 cells were treated with a non-permeable NA inhibitor, Zanamivir. We observed that Zanamivir treatment reduced the proportion of

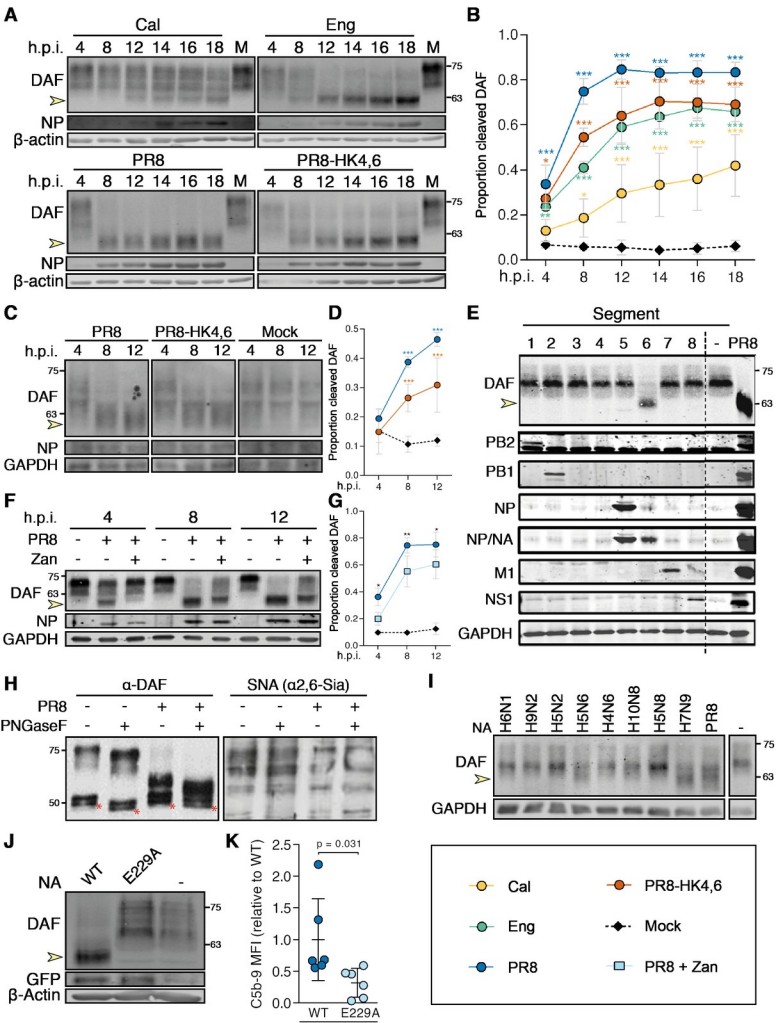

**Fig 10. Influenza A virus neuraminidase cleaves DAF through its sialidase activity. A:** Western blot detection of complement decay-accelerating factor (DAF) in A549 cells upon infection with A/California/7/2009 (Cal), A/England/195/2009 (Eng), A/Puerto Rico/8/1934 (PR8) or A/X-31 (PR8-HK4,6) at a multiplicity of infection (MOI) of 5. **B:** The proportion of cleaved DAF was measured in each lane as the ratio of low molecular weight (MW) to total DAF pixel densitometry. **C:** Western blot detection of DAF in mouse embryonic fibroblasts (MEFs) derived from C57/BL6 WT or *Daf⁻/⁻* mice upon infection with PR8 or PR8-HK4,6 at a MOI of 5. **D:** The proportion of cleaved DAF was measured in each lane as the ratio of low MW to total DAF pixel densitometry. (**B, D**: data shown as mean±sd, from three independent experiments). **E:** Western blot detection of DAF in HEK293T cells after transfection with plasmids encoding the eight different PR8 viral segments. **F:** Western blot detection of DAF in A549 cells upon infection with PR8 at a MOI of 5, treated with Zanamivir. **G:** The proportion of cleaved DAF was measured in each lane as the ratio of low MW to total DAF pixel densitometry (data shown as mean±sd, from four independent experiments). **H:** DAF was purified by immunoprecipitation from cell lysates of A549 cells infected with PR8 at 12 hours post-infection (h.p.i.), treated with PNGaseF and analyzed by western blot or lectin blot with *Sambucus nigra* agglutinin (SNA) (* indicates IgGs from immunoprecipitation). Results are representative of three independent experiments. **I:** Western blot detection of DAF in HEK293T cells after transfection with plasmids encoding NAs from the indicated avian IAVs: H6N1 A/chicken/Taiwan/67/2013, H9N2 A/chicken/Pakistan/UDL-01/08, H5N2 A/goose/Taiwan/01031/2015, H5N6 A/chicken/Jiangxi/02.05 YGYXG023-P/2015, H4N6 A/chicken/Hunan/S1267/2010, H10N8 A/chicken/Jiangxi/1204/2014, H5N8 A/scarlet ibis/Germany/Ar44-L01279/2015, H7N9 A/Anhui/1/2013. **J:** Western blot detection of DAF in HEK293T cells transfected with eight plasmids encoding each of the PR8 segments, including wild-type NA (WT) or the catalytically-impaired mutant NA-E229A (E229A). **K:** Flow cytometry detection of C5b-9 deposition in A549 cells after transduction with WT or catalytically-impaired mutant NA-E229A and treatment with serum (data shown as mean±sd, from six independent experiments, each corresponding to a pool of five independent transductions. Each point represents the median fluorescence intensity (MFI) of a sample treated with serum minus its corresponding heat-inactivated control.). Yellow arrows indicate cleaved DAF. MW is indicated in kDa. Statistical analysis detailed in materials and methods.

cleaved DAF (0.60 vs. 0.75), showing that DAF cleavage happens in part at the cell membrane, and in part in the cytoplasm (Fig 10F and 10G).

For IAV receptor recognition, the binding of sialic acid to the penultimate galactose residues of carbohydrate side chains is important, and different IAVs exhibit preference for Neu5Ac α(2,3)-Gal (hereafter α2,3-) or Neu5Ac α(2,6)-Gal (hereafter α2,6-) conformations [63,64]. Interestingly, most avian IAVs bind preferentially to sialic acid joined to the sugar chain through an α2,3-linkage, whereas human IAV preferentially use α2,6-linked sialic acid as a cellular receptor [64,65]. To assess which type of ligations were cleaved by NA, we infected A549 cells with PR8 and purified DAF by immunoprecipitation. Subsequently, we treated immunoprecipitated DAF with PNGaseF to remove N-glycans, and probed DAF by western blot and lectin blot with *Sambucus nigra* agglutinin (SNA), which detects α2,6-linked sialic acid (Fig 10H). The cumulative effect in DAF MW decrease of PR8 infection and PNGaseF treatment, as well as loss of SNA staining only upon infection, indicates that PR8 infection specifically removes α2,6-linked sialic acid from DAF O-glycans.

The affinity of the IAV HA and NA for respective sialic acid conformation is one of the host species restriction factors [66], avian strains preferring α2,3-linked sialic acids, whereas human strains are able to cleave α2,6-linked sialic acids. In accordance with that, transfection of HEK293T cells with avian-adapted NAs did not impact DAF MW (Fig 10I). Remarkably, transfection with NAs from a H7N9 isolated from a human patient (A/Anhui/1/2013) and from a H5N6 isolated from a chicken (A/chicken/Jiangxi/02.05 YGYXG023-P/2015) caused a drop in DAF MW. These two NAs are thus able cleave α2,6-linked sialic acid residues, indicating they are already adapted to human sialic acid linkages and indeed both H7N9 [67,68] and H5N6 [69,70] strains have been shown to cause severe zoonotic disease. These results suggest that analysis of sialic acid cleavage might be worth exploring as a measure for host adaptation and zoonotic events.

NA unprecedented direct and pronounced effect on DAF strongly suggests a functional consequence. It has been proposed that DAF negatively charged sialic acids function as a spacer, which projects DAF RCA domains to the extracellular milieu [71]. Additionally, sialic acid removal promotes DAF to be proteolytically shed [35]. Therefore, we hypothesized that NA-mediated sialic acid cleavage would result in DAF loss/alteration of function, resulting in increased complement activity. To validate this hypothesis, we aimed at engineering recombinant mutant viruses composed of seven PR8 segments (segments 1–5, 7–8) and expressing segment 6 from viruses that do not cleave DAF. If indeed our hypothesis is correct, the absence of DAF cleavage would supress the protective effect observed in *Daf*[-/-] mice. Supported by data in Fig 10I, we selected segment 6 from H5N2 and H5N8 virus to produce RG 7+1 PR8 reassortant viruses (S5A Fig). After transfecting HEK293T cells with the eight plasmids to produce the P0 viruses, we observed that PR8 NA-H5N2 did not cleave DAF, as expected (S5B Fig). However, PR8 NA-H5N8 viruses acquired the capacity to cleave DAF (S5B Fig). For this reason, even though we amplified both viruses in eggs to avoid additional adaptation to host environment (S5C Fig), we excluded the PR8 NA-H5N8 from further experiments as they had acquired the ability to cleave DAF. The rescued PR8 NA-H5N2 in eggs (S5C Fig), was then tested for its ability to infect and replicate in A549 cells. One-step infection at high MOI resulted in effective replication, though PR8 NA-H5N2 grew to a significantly lower extent than PR8 (S5D Fig). Remarkably, when A549 cells were infected, PR8 NA-H5N2 cleaved DAF (S5E Fig). These results strongly suggest that IAVs containing NAs from avian origin rapidly adapt to cleave DAF α2,6-linked sialic acid, and could not be used to assess the link between DAF cleavage *in vitro* and pathogenicity upon mouse infection.

Hence, to assess the effect of DAF sialic acid cleavage in complement activation and to overcome these mechanisms of fast adaptation, we opted for a completely *in vitro* approach. We

introduced the mutation E229A in PR8 segment 6, which pronouncedly decreases NA enzymatic activity, while still sustaining a low level of viral replication [72]. We then produced lentiviral vectors to deliver WT or E229A versions of PR8 NA fused to GFP, and confirmed that NA-E229A had impaired DAF cleavage (Fig 10J). After transduction of A549 cells, we treated cells with normal human serum and stained for C5b-9 as a proxy for complement activation. Transduction of cells with WT NA resulted in increased C5b-9 deposition when compared with cells transduced with E229A (1±0.7 vs. 0.3±0.2) (Fig 10K). Therefore, NA removal of DAF sialic acid content does impair its complement regulator function, increasing complement activation.

Overall, our results unveil DAF as a novel host virulence factor upon IAV infection, depending on interaction with HA and NA. Specifically, we observed a widespread direct interaction between NA and DAF with functional implications, which is an unprecedented way of a virus, via altering a host protein from within the infected cell, modulating the immune response.

## Discussion

This work highlights the importance of a balanced immune response to viral infections in order to clear the disease without causing immunopathology. Despite its intrinsic protective role, complement is a documented driver of immunopathology in severe viral infections such as IAV [31–33], SARS-CoV-2 [28–30] and MERS [27]. In the context of IAV, inhibition of different components of the complement system such as C3a receptor and C5 decreased immune cell recruitment and activation leading to an ameliorated disease outcome [31–33]. Our work is in accordance with these studies as *Daf*$^{-/-}$ mice have less severe disease upon IAV infection, coupled with reduced C3a levels in BALs, viral immune response and number of immune cells recruited to the lungs (Figs 4B, 5, 7B and 9B). However, C3 is essential in IAV infection. *C3*$^{-/-}$ and *C3*$^{-/-}$ / *Daf*$^{-/-}$ mice had increased weight loss when compared to the WT (Figs 4A, 7A and 9A), and *C3*$^{-/-}$ mice presented increased lung inflammation and infiltration of immune cells upon IAV infection [47,73]. These observations show the potential of regulating complement activation as a strategy to provide resilience to viral infections, without affecting pathogen clearance.

Interestingly, infection of *Cd59*$^{-/-}$ mice and analysis of CDC in WT and *Daf*$^{-/-}$ primary lung cells indicated that the last step of the complement cascade does not impact disease outcome in IAV infection (Figs 2E, 2F, 4C, S4F and S4G). Rather, it suggests that earlier components of the complement cascade, such as anaphylatoxins C3a and/or C5a have a modulatory role of IAV virulence. This hypothesis agrees with the function of C3a and C5a as recruiters and activators of the innate immune response, which can lead to immunopathology [31–33]. Our results indicate that, in fact, and contrary to expected, in IAV infection lack of DAF leads to reduced activation of complement, lower levels of C3a and viral response chemokines, and decreased recruitment of monocytes and neutrophils, specifically (Figs 4B, 4E, 4F and 5). The lower levels of C3a, CCL2, CXCL1, CCL1 and CXCL10 detected in the BALs of *Daf*$^{-/-}$ mice could explain the lower numbers of innate immune cells recruited, and decreased tissue damage. Interestingly, DAF depletion does not ameliorate disease outcome in all IAV infections. We observed that although *Daf*$^{-/-}$ mice were protected in Eng, Cal, PR8-HK4,6 and PR8-HK4 infections, levels of mortality were identical in WT and *Daf*$^{-/-}$ mice when infected with PR8 and PR8-HK6. These differences could be due to the levels of virulence, PR8 and PR8-HK6 being more virulent than the other strains. However, we have also shown that compared to PR8-HK4,6, infection with PR8-HK6 altered recruitment of adaptive immune cells, and PR8-HK4 of innate immune cells, without changing the levels of C3a in *Daf*$^{-/-}$ mice. These

results indicate that complement is not the sole recruiter and activator of the immune response, and that a direct or indirect HA-DAF and/or NA-DAF interaction has additional roles to play in immune cell recruitment. Therefore, interaction between DAF and different HA and NA could elicit different immune responses. Indeed, although PR8, Eng and Cal are all H1N1 viruses, the corresponding HA and NA proteins are different and may be translated in differences in virulence as seen in [45]. Further analysis of PR8, Eng and Cal infections in *Daf*$^{/-}$ mice would allow clarification of the impact of different HA-DAF and NA-DAF interactions.

In fact, we found that HA-DAF interplay impacts recruitment of CD4$^+$ and CD8$^+$ T cells, both of which shown to be essential in the clearance of IAV [74]. The lower levels of these cells in *Daf*$^{/-}$ mice might annul the beneficial effect of lower lung tissue damage observed at 6 d.p.i.. Indeed, upon PR8-HK6 challenge, mice bodyweight rapidly dropped at 7 d.p.i., whereas in infection with other viral strains loss of weight started around 4 d.p.i. and was more gradual, suggesting that the adaptive immune system is implicated in the process [9,75]. Despite HA being amongst the most immunogenic proteins of IAV, and hence its involvement in adaptive immune response not surprising [75,76], our work shows for the first time a specific interaction of HA with DAF and the implications of this axis in T cell recruitment. Analysis of T cell chemoattractants such as CCL5 and CXCL10 would help clarifying this question.

We also identified a novel function for the viral protein NA, via cleaving sialic acids of DAF and modulating immune cell recruitment and viral pathogenicity. Remarkably, NA-mediated cleavage of another host protein, latent TGF-β, activates it, which confers a protective role upon infection [77,78]. Here we found that NA cleaves α2,6-linked sialic acids from DAF and hypothesize that this could increase viral immune response and explain the differences in the numbers of neutrophils and monocytes recruited to the site of infection. Ablation of neutrophils in IAV infections have been shown to prevent tissue damage without affecting viral loads [79–82]. In fact, these cells have long been associated with acute respiratory distress syndrome [83], and extensive neutrophil infiltration and release of neutrophil extracellular traps (NETs) have been linked to increased pneumonia severity in critical cases of COVID-19 [84–86]. Despite these observations, neutrophils are important to the host response against IAV infection as neutrophil depletion resulted in exacerbated viral loads, lung damage and mortality in mice infected with PR8-HK4,6 [87,88]. In addition to neutrophils, monocytes are readily recruited to sites of IAV challenge where they differentiate into macrophages or dendritic cells (DC) [89,90] that share many properties with their conventional counterparts [91] and have been studied upon IAV infection [91,92]. Monocyte-derived macrophages contribute to the inflammation resolution by clearing apoptotic neutrophils and confer lasting protection against secondary bacterial infections [92,93]. The interaction with apoptotic neutrophils has also been reported to increase differentiation of monocytes into DC, promoting adherence of CD8$^+$ T cells [93]. Conversely, monocyte and monocyte-derived cells may contribute to immunopathology, as their depletion decreased disease severity without altering viral loads [94–96]. These studies show that both cell types are essential for IAV infection but can contribute to tissue damage, and support our hypothesis that increased immunopathology of WT mice upon IAV infection is mediated by excessive recruitment of neutrophils and monocytes.

The link we identified via NA, DAF and complement establishes a viral mediated mechanism for activating inflammation via increasing the recruitment of immune cells. The model that we propose and that is depicted in Fig 11 explores an interplay between HA and NA in modulating the immune response. Previous examples include the activation of the NK cell sialylated receptors NKp44 and NKp46 by HA at the surface of infected cells, which is countered by NA-mediated desialylation [97,98]. In the case of our work, it is known that apical delivery of NA to the cell surface is potentiated by HA [99] and during this transport (and also at the

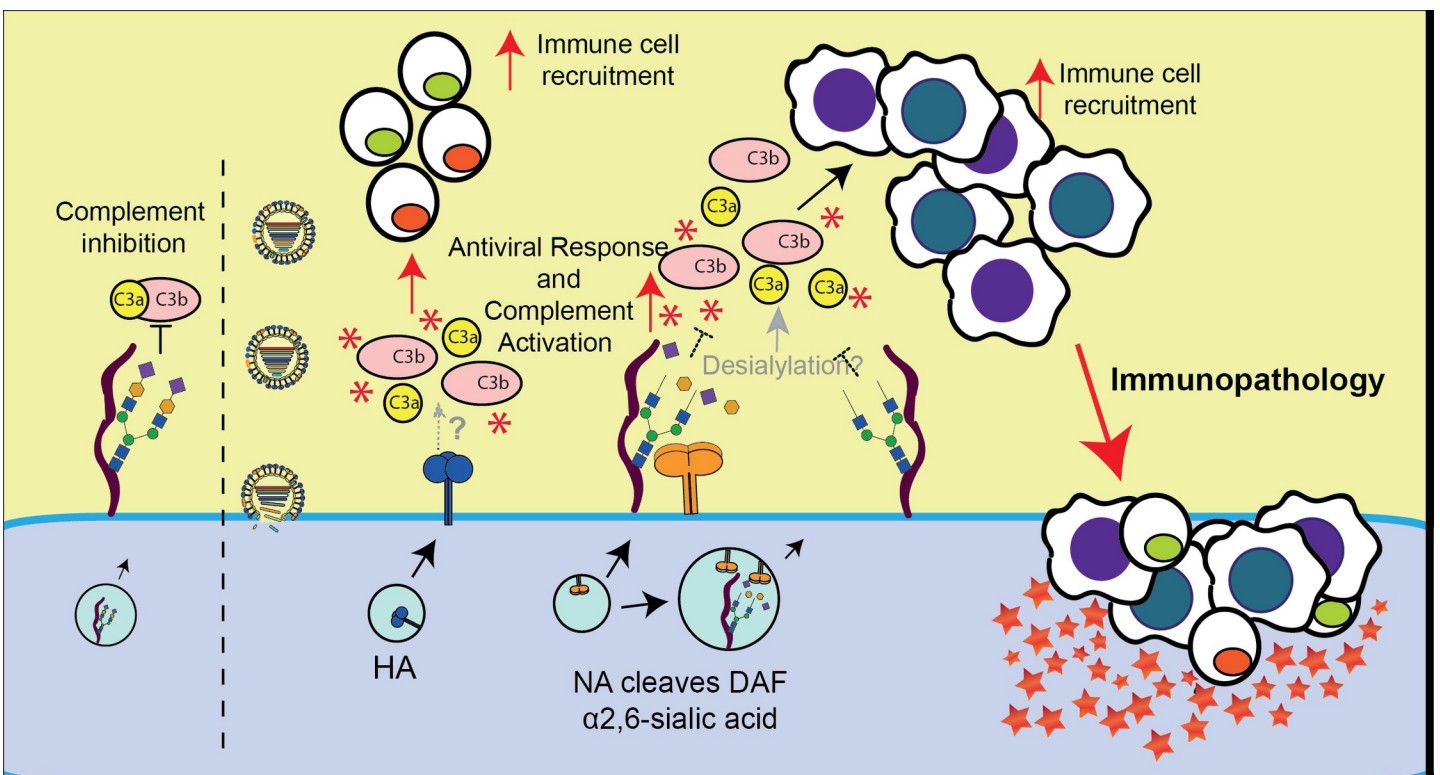

**Fig 11. Proposed model for DAF-mediated immunopathology.** At steady state, DAF accelerates the decay of C3 convertases, inhibiting the formation of C3a and C3b and subsequent complement activation. Upon IAV infection, the cell will produce viral proteins, and in particular HA (in blue) and NA (in orange). NA is a potent sialidase that will remove the sialic acid content of DAF both in the cytoplasm and at the surface. This processing of DAF by NA may lead to DAF loss/alteration of function. Desialylation of DAF would then impair the inhibition of C3 convertases leading to overactivation of the complement pathway. In parallel, the presence of HA at the cell surface may exacerbate complement activation that could then not be controlled by a desialylated DAF. The activation of complement pathway via HA, combined with the lack of inhibition due to NA sialidase activity would induce complement overactivation, leading to C3a release. Moreover, the presence of HA at the cell membrane and recognition of desialylated surface proteins by PRR would induce an exacerbated immune response (red asterisks). Consequently, an excess of innate immune cell recruitment and inflammation would promote tissue damage and ultimately immunopathology, worsening disease outcome. (Black arrows indicate confirmed events whereas grey arrows indicate speculative events needing further validation. Red arrows indicate events that will worsen disease outcome).

plasma membrane), NA would cleave DAF sialic acid giving rise to increased activation of complement. Indeed, we observed that IAV infection induces a drop in DAF MW over the course of infection both in human and murine cell lines. The drop corresponds to DAF sialic acid content, and NA is necessary and sufficient for this cleavage (Fig 10A, 10B, 10E and 10J). Moreover, transduction of cells with a functional NA and thus removal of DAF sialic acid content resulted in an increased C5b-9 deposition (Fig 10K). We propose that the removal of DAF sialic acid content would not lead to a loss of function, but instead trigger an exaggerated complement response. This is contrary to what is observed for autoimmune diseases, for which $Daf^{-/-}$ mice have been widely used [100–102]. These mice have increased disease severity coupled with high complement activation levels when compared to their WT counterparts, showing that $Daf^{-/-}$ mice do not lack the ability to activate the complement and that the mechanism we now describe could be shared among viruses containing promiscuous NAs. As an alternative, NA-mediated DAF cleavage could result in the recruitment of innate immune cells by exposing "non-self" glycans at cell surface, which has been shown to activate complement via the lectin pathway [103]. Besides complement, it could also be recognized by different PRRs and induce the observed increase in chemokines [104]. At the moment this hypothesis is speculative, but raises concerns about using therapies, such as DAS181 [105], aiming at decreasing

sialic acid levels at cell surface to prevent viral entry. Interestingly, our work indicates that NA cleavage of sialic acids does not happen solely at the cell surface, but also in the cytoplasm, as treatment with Zanamivir did not completely abolish DAF cleavage (Fig 10F). To the best of our knowledge, this mechanism has not been reported before.

DAF cleavage provides a possible link between DAF-NA interaction and *in vivo* pathology. Given that our study shows that sialic acids cleaved by DAF are α2,6-linked to O-glycans, this mechanism may have implications in host species jumps, as for example, IAV adapted to birds exhibit preference for α2,3-linked sialic acids. Interestingly, we present evidence that NAs derived from two avian-adapted strains, H5N6 and H7N9, were able to cleave human DAF (Fig 10I). As H7N9 and H5N6 outbreaks provoked severe infections in humans, associated with exacerbated immune response [67–69], hypothetically establishing DAF cleavage as a hallmark of virulence could be a useful tool to monitor viruses with pandemic potential.

In addition, many host proteins including mucins are decorated by sialic acids. Mucins form an important barrier at the cell surface preventing viral entry [106]. These proteins are also heavily glycosylated, specifically at the terminal part of O-glycans [107], similarly to DAF, indicating that they could be substrates of NA. As a consequence, the mechanism we describe could be used to manipulate the extracellular environment and facilitate viral cell-to-cell transmission. Identification of glycans exposed at the surface of infected cells and their interaction with viral proteins may help understand the balance between viral entry and immune response targets and reveal disease resilience pathways prone to therapeutic intervention.

## Materials & methods

### Ethics statement

All experiments involving mice were performed using 8-week-old littermate C57BL6/6J, C57BL6/6J *Daf*$^{-/-}$, C57BL6/6J *Cd59*$^{-/-}$, C57BL6/6J *C3*$^{-/-}$ or C57BL6/6J *C3*$^{-/-}$ / *Daf*$^{-/-}$ female mice under specific pathogen-free conditions at the Instituto Gulbenkian de Ciência (IGC) biosafety level 2 (BSL-2) animal facility. Animals were group housed in individually ventilated cages with access to food and water *ad libitum*. This research project was ethically reviewed and approved by both the Ethics Committee and the Animal Welfare Body of the IGC (license references: A016/2013 and A013/2019), and by the Portuguese National Entity that regulates the use of laboratory animals (DGAV–Direção Geral de Alimentação e Veterinária (license references: 0421/000/000/2015 and 0421/000/000/2020). All experiments conducted on animals followed the Portuguese (Decreto-Lei n˚ 113/2013) and European (Directive 2010/63/EU) legislations, concerning housing, husbandry and animal welfare.

### Statistical analyses

All statistical analyses were conducted using GraphPad Prism 6. Detailed statistics and number of replicates for all experiments can be found in the figure legends and/or in the manuscript. Bodyweight loss and DAF cleavage: Statistical significance represented as $^*p < 0.05$, $^{**}p < 0.01$, $^{***}p < 0.001$, using two-way ANOVA followed by Holm-Sidak multiple comparisons test. Survival curves: Statistical significance compared with WT using Log-rank (Mantel-Cox) test. Compare two groups: Population normality assessed with D'Agostino & Pearson omnibus normality test. Statistical significance using unpaired t-test with Welch's correction for normal populations or Mann-Whitney test for populations whose normality was not proved; Wilcoxon matched-pairs signed rank test for populations whose normality was not proved, but samples were paired. Multiple comparisons: Population normality assessed with D'Agostino & Pearson omnibus normality test. Kruskal-Wallis followed by Dunn's multiple comparisons test for populations whose normality was not proved.

### Mice infection, lung, BAL and tissue analysis

All mice experiments were conducted in a BSL-2 animal facility. Littermates were randomly allocated to experimental conditions. Mice were anesthetized with isoflurane (Abbot) and 30μL of inoculum administered intranasally. Mice were daily monitored for 11 days or sacrificed with $CO_2$ at indicated timepoints. Animals that presented a body weight loss of more than 25% were closely monitored. To comply with best animal welfare practices, animals that maintained a reduction of initial bodyweight of more than 25% for 24h were sacrificed. However, animals that regained enough weight to be above the -25% threshold after those same 24h were allowed to recover. Tissues were collected in aseptic conditions.

Lung viral loads were collected from right lower lobes using tungsten carbide beads (Qiagen) in a TissueLyser II (Qiagen) at $20s^{-1}$ for 3min. After centrifugation, supernatants were collected and titrated by plaque assay [108,109].

Bronchoalveolar lavage (BAL) of the whole lung was performed with 1mL sterile PBS via tracheal cannula. After centrifugation, supernatants were used for ELISA analysis (Mouse C3a (TECOmedical, TE1038) or Mouse IFN-γ DuoSet ELISA (R&D Systems, DY485)) and multiplex cytokine detection (LEGENDplex Mouse Anti-Virus Response Panel (BioLegend, 740621)) following manufacturer's instructions, and cells analyzed by flow cytometry. Unspecific staining was minimized with Fc blocking (rat anti-mouse CD16/CD32, IGC antibody facility, clone 2.4G2). Cells were incubated with primary antibodies (S2 Table) in FC buffer, 20min at 4°C, stained with Zombie Aqua Fixable Viability Kit (BioLegend, 423101) and fixed with IC fixation buffer according to manufacturer's recommendations. Flow cytometry analysis of cell populations was performed in a BD LSR Fortessa X-20 SORP (BD Biosciences) equipped with BD FACSDiva 8 and FlowJo 10 software (Tree Star Inc., Ashland, OR, USA), and absolute numbers obtained with Perfect-Count Microspheres (Cytognos, CYT-PCM). Representative gatings are illustrated in S6 Fig.

### Immune cell depletion

Natural killer (NK) cells were depleted by intraperitoneal (IP) injection of 200μg α-NK1.1 (IGC antibody facility, clone PK136) in 200μL PBS every 72h, starting 72h before infection.

### Histology and immunohistochemistry (IHC)

Histological scoring was conducted as in [45], and expressed as the sum of the parameter described in S1 Table. Scoring was performed blindly by a pathologist. For IHC tissue sections were deparaffinized, rehydrated, and heated in citrate buffer (40mM sodium citrate dihydrate, 60mM citric acid, pH 6) and blocked with 1:50 Fc block reagent (rat anti-mouse CD16/CD32, IGC antibody facility, clone 2.4G2). Slides were then incubated with rabbit α-NP (102) 1:1000 for 16h at 4°C. After blocking of endogenous peroxidases sections were incubated with Imm-PRESS HRP Horse Anti-Rabbit IgG Polymer Detection Kit (Vector Laboratories, MP-7401-15) for 1h at RT and then with DAB substrate (Roche, 11718096001) according to manufacturer's instructions. Finally, lung sections were contrasted with Mayer Hematoxylin and images taken in a NanoZoomer-SQ Digital slide scanner (Hamamatsu Photonics).

Cell infiltrates in lung parenchyma were assessed semi-quantitatively by adding scores obtained for alveolar inflammation, interstitial inflammation and perivascular/peribronchiolar inflammation.

### Complement dependent toxicity and C5b-9 deposition

Lung primary cells from WT or $Daf^{-/-}$ mice were infected as described below with indicated IAVs for 12h, collected, and suspended in veronal buffer (CompTech, B100). Serum from WT

mice (or heat-inactivated at 56˚C, 30min) was added at a final concentration of 50% (v/v) and incubated at 37˚C, 5% $CO_2$ for 1h. Viability was assessed by flow cytometry using Zombie Aqua Fixable Viability Kit after fixation IC fixation buffer following manufacturer's indications.

C5b-9 deposition measurement was adapted from [110]. Lentivirus encoding NA-GFP WT/E229A was added to A549 cells. Cells were then suspended in veronal buffer and human serum (Sigma-Aldrich, H4522) (or heat-inactivated for 30min at 56˚C), added at a final concentration of 50% (v/v). After 15min incubation, C5b-9 deposition was assessed by flow cytometry using α-C5b-9 (Abcam, ab55811, 1:100) and α-Mouse Alexa Fluor 647 (Invitrogen, A31571, 1:1000). Cells were fixed with IC Fixation Buffer according to manufacturer's indications.

## Cell lines, transfections and infection

Madin-Darby Canine Kidney (MDCK), Human Embryonic Kidney 293 T (HEK293T), and human alveolar basal (A549) cells were a kind gift from Prof. Paul Digard (Roslin Institute, UK). Mouse embryonic fibroblasts (MEFs) were isolated from WT and $Daf^{-/-}$ mice E13.5 to E15.5 embryos as previously described [111]. Primary lung cells were isolated from WT and $Daf^{-/-}$ mice. Briefly, 1.5mL of sterile collagenase D (0.5mg/mL in PBS, Roche, 11088858001) and 0.5mL of melted agarose (1% in PBS, Lonza, 50004) were instilled in lungs of mice after exsanguination and PBS perfusion. Whole lungs were then collected and incubated with collagenase D 40min at RT. After dissection in complete DMEM supplemented with 5U DNase I (NZYTech, MB19901), cells were collected and plated in a 6-well plate at a density of 9 x $10^5$ cells/well and incubated for 48h at 37˚C, 5% $CO_2$. All cell lines were cultured in complete DMEM and incubated at 37˚C, 5% $CO_2$.

Transfection of HEK293T cells was performed using Lipofectamine 2000 (ThermoFisher, 11668027) according to manufacturer's recommendations. Plasmids encoding NA genes from following strains were kindly provided by Dr. Holly Shelton (The Pirbright Institute, UK) and were synthesized by GeneArt (Invitrogen) and cloned into a pHW2000 vector [112]: H6N1 A/chicken/Taiwan/67/2013 (GenBank accession no. KJ162862), H9N2 A/chicken/Pakistan/UDL-01/08 [113], H5N2 A/goose/Taiwan/01031/2015 [114], H5N6 A/chicken/Jiangxi/02.05 YGYXG023-P/2015 [114], H4N6 A/chicken/Hunan/S1267/2010 (GenBank accession no. KU160821), H10N8 A/chicken/Jiangxi/1204/2014 (GenBank accession no. KP285359), H5N8 A/scarlet ibis/Germany/Ar44-L01279/2015 [114], H7N9 A/Anhui/1/2013 [115].

One-step infections were carried out at a multiplicity of infection (MOI) of 3 in serum-free DMEM for 45min and then overlaid with complete DMEM and kept at 37˚C and 5% $CO_2$ for the duration of the experiment.

## Western blot

Western blotting was performed according to standard procedures and imaged using a LI-COR Biosciences Odyssey Infrared Imaging System. Primary and secondary antibodies used are in S2 Table.

## Viruses and titration

Human circulating strains A/California/7/2009 (Cal, H1N1) and A/England/195/2009 (Eng, H1N1) were kindly provided by Prof. Paul Digard). Reverse-genetics derived A/Puerto Rico/8/34 (PR8) and A/X-31 (PR8-HK4,6) were used as model viruses. Reverse-genetics derived chimeric PR8 containing the segment 4 from A/Hong Kong/1/1968, seg4-HK68 (PR8-HK4), or the segment 6 (PR8-HK6) were produced as previously described [105,109,110]. pDual

plasmids were a kind gift from Dr. Ron Fouchier (Erasmus MC, Netherlands). PR8 NA-E229A [55] was generated by reverse genetics after site directed mutagenesis of pDual:: segment6. All viruses were amplified in embryonated chicken eggs and titrated using plaque assay as previously described [100,101].

Lentivirus were produced in HEK293T cells transfected with the following plasmids (ThermoFisher, OHS4735): 6μg pLEX-MCS-1::NA-GFP WT/E229A, 4.2μg psPAX2, 1.8μg pMD2.G. 72h hours after transfection, medium containing lentivirus was collected and stored at -80°C.

## Bacteria and cloning

All transformations for cloning or plasmid amplification were performed in *Escherichia coli* XL10 Gold (Agilent) according to manufacturer's instructions.

Viral RNA (vRNA) was extracted from egg-grown viral stocks using QIAamp Viral RNA Mini Kit (Qiagen, 50952904) according to manufacturer's instructions. From purified vRNA, NA cDNA was produced using NZY M-MulV First-Strand cDNA Synthesis Kit (NZYTech, MB17302) with primer "NA_Fw_HindIII" following manufacturer's recommendations. To produce pEGFP-N1::NA, NA was then amplified and cloned in HindIII-KpnI restriction sites of pEGFP-N1. To generate pLEX-MCS-1::NA-GFP, NA-GFP was amplified from pEGFP-N1:: NA and cloned into NotI/XhoI sites of pLEX-MCS-1. pDual::seg6-E229A and pEGFP-N1:: NA-E229A were generated by site directed mutagenesis of pDual::seg6 and pEGFP-N1::NA respectively, using the QuikChange Site-Directed Mutagenesis Kit (Agilent, 200518), according to manufacturer's instructions. Primer sequences are indicated in S3 Table.

## DAF glycosylation

A549 cells were infected with PR8 as described above. After 12h of infection, cells were lysed with lysis buffer 17 (R&D Systems, 895943) and protein quantified using bicinchoninic acid protein assay (BCA) (Pierce 23225). Protein G Sepharose 4 Fast flow beads (GE Healthcare, GE17-0618-01) were incubated with α-DAF (Abcam, ab133684) for 5h and then Protein G-DAF complexes were crosslinked using bis(sulfosuccinimidyl)suberate ($BS^3$) (Sigma, S5799). Protein from total cell extracts (100 μg) were then added to the antibody Protein G complex and incubated 16h at 4°C in a rotator mixer. Washing steps were performed with PBS and samples used for downstream analysis.

After DAF immunoprecipitation, removal of N-glycans was performed by digestion with PNGaseF (New England Biolabs, P0704S), according to manufacturer's instructions. For blotting experiments gels were transferred onto nitrocellulose and unspecific binding blocked using 5% BSA and 2% polyvinylpyrrolidone (PVP) for blot detection with α-DAF or biotinylated *Sambucus nigra* agglutinin (SNA) (Vector Laboratories B-1305-2), respectively. DAF was detected with HRP-conjugated goat anti-rabbit (Jackson ImmunoResearch, 111-035-144) and SNA with Vectastain Avidin/Biotin Complex (Vector Laboratories, PK-4000) incubation. Detection was performed by enhanced chemiluminescence (ECL) (GE Healthcare, RPN2232) and film sheet exposure.

## Supporting information

**S1 Fig. Representative histological sections of *Daf*$^{-/-}$ and WT mice infected with PR8-HK4,6, PR8-HK6, and PR8-HK4.** C57BL/6J WT and *Daf*$^{-/-}$ mice were mock infected (A) or infected with 1000 PFU of PR8-HK4,6 (B), 20 PFU of PR8-HK6 (C) and 100 PFU of PR8-HK4 (D). Samples were collected at the indicated time points. (TIF)

**S2 Fig. NK cell depletion does not alter disease outcome. A:** Representative flow cytometry detection of NK cells (gated in CD45$^+$ population) in C57BL/6J WT 72 hours after depletion via intraperitoneal (IP) injection of α-NK1.1. **B:** Bodyweight loss of C57BL/6J WT mice infected with 100 PFU of A/X-31 (PR8-HK4,6) and depleted of NK cells by IP injection of α-NK1.1 every 72 hours, starting 72 hours before infection (Inf n = 5 and mock n = 1 per group). Results are expressed as mean±sd. Statistical analysis detailed in materials and methods. (TIF)

**S3 Fig. Absolute values of immune cell recruitment in BALs in representative experiments. A**, **B**, **C, D**: Analysis of neutrophils (**A**) monocytes (**B**) CD4$^+$ T cells (**C**) and CD8$^+$ T (**D**) cells levels in BALs of WT or *Daf$^{/-}$* mice infected with 1000 PFU A/X-31 (PR8-HK4,6). Samples were collected at 3 d.p.i. (n = 4 per group) and 6 d.p.i. (n = 5 per group) and 9 d.p.i. (n = 5 per group). **E, F, G, H**: Analysis of neutrophils (**E**) monocytes (**F**) CD4$^+$ T cells (**G**) and CD8$^+$ T (**H**) cells levels in BALs of WT or *Daf$^{/-}$* mice infected with 1000 PFU PR8-HK6. Samples were collected at 3 d.p.i. (n = 5 and n = 6 for WT and *Daf$^{/-}$* respectively), and 6 d.p.i. (n = 5 per group). **I, J, K, L**: Analysis of neutrophils (**I**) monocytes (**J**) CD4$^+$ T cells (**K**) and CD8$^+$ T (**L**) cells levels in BALs of WT or *Daf$^{/-}$* mice infected with 1000 PFU PR8-HK4. Samples were collected at 3 d.p.i. (n = 6 and n = 5 for WT and *Daf$^{/-}$* respectively) and 6 d.p.i. (n = 4 and n = 5 for WT and *Daf$^{/-}$* respectively). Samples with non detectable cell levels are noted as n.d. (TIF)

**S4 Fig. DAF does not affect replication of PR8, PR8-HK4,6, PR8-HK4 and PR8-HK6. A:** Measurement of viral plaques diameter after infection of MDCK cells monolayers. Data shown as mean±sd from two independent experiments, each corresponding to six independent infections for each virus. Each point represents an individual plaque. **B-E:** Replication kinetics of A/Puerto Rico/8/1934 (PR8) **(B)**, A/X-31 (PR8-HK4,6) **(C)**, PR8 containing the segment 4 of A/Hong Kong/1/68 (HK68) (PR8-HK4) **(D)** and PR8 containing the segment 6 of HK68 (PR8-HK6) **(E)** in mouse embryonic fibroblasts (MEFs) derived from C57BL/6J WT or *Daf$^{/-}$* mice at multiplicity of infection (MOI) = 0.005. Data shown as mean±SEM, from two independent experiments. Statistical analysis detailed in materials and methods. **F, G**: Cell death of primary lung cells derived from WT or *Daf$^{/-}$* mice infected or mock-infected with PR8, PR8-HK4,6, PR8-HK4 or PR8-HK6 and treated with serum **(F)** or heat-inactivated control **(G)**. Results are expressed as mean±sd from three replicates from two independent experiments. Statistical analysis detailed in materials and methods. (TIF)

**S5 Fig. Influenza A viruses with avian NA promptly adapt to cleave DAF. A:** As showed in Fig 10I, HEK293T cells were transfected with eight different avian derived NAs. **B:** To produce reverse genetics (RG) reassortant A/Puerto Rico/8/34 (PR8) viruses, HEK293T cells were transfected with seven plasmids encoding segments 1–5, 7 and 8 from PR8, and the segment 6, which encodes NA, from the indicated viruses. **C:** After one round of amplification in embryonated chicken eggs, recovered viruses were titrated. **D-E:** A549 cells were infected with PR8 NA-H5N2 at multiplicity of infection (MOI) of 3 and samples collected at the indicated timepoints to titrate released virions (**D**) and analyze DAF cleavage by western blot (**E**) (**D**: pooled data from three independent experiments; **E**: representative blot from three independent experiments). Statistical analysis detailed in materials and methods. (TIF)

**S6 Fig. Representative flow cytometry gating strategy.** (TIF)

**S1 Table. Histological scoring parameters.**
(TIF)

**S2 Table. Antibodies used in flow cytometry and western blot.**
(TIF)

**S3 Table. Primers used for cloning and site-directed mutagenesis.**
(TIF)

## Acknowledgments

We acknowledge Dr. Colin Adrain (IGC, Portugal), Dr. Holly Shelton (The Pirbright Institute, UK), Dr. Jonathan Yewdell (NIAID, USA), Dr. Luís Moita (IGC, Portugal), Dr. Miguel Soares (IGC, Portugal), Prof. Paul Digard (Roslin Institute, UK), Dr. Ron Fouchier (Erasmus, Netherlands), Dr. Vera Martins and Prof. Wen-Chao Song (University of Pennsylvania, USA) for providing mice, cells and reagents. We are grateful to the Animal House Facility, Flow Cytometry Facility and Histopathology Unit at the IGC for technical support, sample processing and data collection. We thank André Barros (IGC Portugal), Dr. Mónica Bettencourt-Dias (IGC, Portugal), Dr. Gabriel Nuñez (University of Michigan) and the members of CBV lab for helpful discussion.

## Author Contributions

**Conceptualization:** Nuno Brito Santos, Zoé Enderlin Vaz da Silva, Maria João Amorim.

**Data curation:** Nuno Brito Santos, Zoé Enderlin Vaz da Silva.

**Formal analysis:** Nuno Brito Santos, Zoé Enderlin Vaz da Silva, Catarina Gomes, Celso A. Reis, Maria João Amorim.

**Funding acquisition:** Maria João Amorim.

**Investigation:** Nuno Brito Santos, Zoé Enderlin Vaz da Silva, Maria João Amorim.

**Methodology:** Nuno Brito Santos, Zoé Enderlin Vaz da Silva, Catarina Gomes, Celso A. Reis, Maria João Amorim.

**Project administration:** Maria João Amorim.

**Resources:** Maria João Amorim.

**Supervision:** Maria João Amorim.

**Validation:** Nuno Brito Santos, Zoé Enderlin Vaz da Silva.

**Visualization:** Nuno Brito Santos, Zoé Enderlin Vaz da Silva, Maria João Amorim.

**Writing – original draft:** Nuno Brito Santos, Zoé Enderlin Vaz da Silva, Maria João Amorim.

**Writing – review & editing:** Nuno Brito Santos, Zoé Enderlin Vaz da Silva, Catarina Gomes, Celso A. Reis, Maria João Amorim.

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
