## [Decision Letter · Decision Letter 0]

10 Mar 2021

Dear Dr. Amorim,

Thank you very much for submitting your manuscript "Complement Decay-Accelerating Factor is a modulator of influenza A virus lung immunopathology" for consideration at PLOS Pathogens. As with all papers reviewed by the journal, your manuscript was reviewed by members of the editorial board and by several independent reviewers. In light of the reviews (below this email), we would like to invite the resubmission of a significantly-revised version that takes into account the reviewers' comments. Please make sure to respond to all comments by the reviewers and include the requested relevant experiments that will strengthen the conclusions. 

We cannot make any decision about publication until we have seen the revised manuscript and your response to the reviewers' comments. Your revised manuscript is also likely to be sent to reviewers for further evaluation.

Sincerely,

Carolina B. Lopez, Ph.D.

Section Editor

PLOS Pathogens

Carolina Lopez

Section Editor

PLOS Pathogens

Kasturi Haldar

Editor-in-Chief

PLOS Pathogens

orcid.org/0000-0001-5065-158X

Michael Malim

Editor-in-Chief

PLOS Pathogens

orcid.org/0000-0002-7699-2064

Reviewer's Responses to Questions

**Part I - Summary**

Reviewer #1: The paper by Santos et al., investigates the role of DAF (complement decay accelerating factor) in influenza virus infection. While initially speculated that a loss of this negative regulator of complement would results in increased complement activation, interestingly the group report that post influenza infection DAF KO mice have lower levels of complement activation, reduced innate cell infiltration and improved disease outcomes.

It was proposed that DAF increases complement activation in IAV infection, increasing immunopathology by enhancing neutrophil and monocyte recruitment but not impacting viral loads. Mechanistically the group show that the interaction of specific viral HA and NA with DAF appear to impact innate and adaptive immune cell activation thereby impacting viral control and tissue damage.

While this is an interesting study, a more comprehensive analysis is required as a lot of the experiments are limited to a single time point, or single inflammatory agent, or single tissue and it is difficult to make conclusions about differences in protection and tissue damage based on such a narrow snapshot of the disease.

Reviewer #2: Complement Decay-Accelerating Factor is a modulator of influenza A virus lung immunopathology

PPATHOGENS-D-21-00327 by Santos et al.

This is a very comprehensive and well-written manuscript reporting on the role of DAF on influenza virus induced immunopathology. Using a series of different influenza virus isolates the investigators noticed a virus strain dependent effect of DAF on weight loss and survival after infection. The most striking difference was found between PR8 and X31, an isogenic set of viruses that differ only in their HA and NA protein expression. Compared to WT mice, X31 infection of DAF-knockout animals resulted in reduced weight loss and mortality, and this was associated with lower concentrations of C3a, reduced numbers of innate immune cells 3 and 6 dpi, and lower histology scores. In contrast, infection of WT or DAF-knockout mice with PR8 resulted in the same amount of weight loss and death after infection suggesting that the HA and NA protein modulate the role of DAF during influenza disease. The C3a concentration and infiltration of innate and adaptive immune cells after PR8 infection were not measured. Next, the authors show that the diminution of inflammation in the absence of DAF is dependent on the expression of C3.

Part two of the manuscript is dedicated to discovering which influenza protein, HA or NA is responsible for this DAF-dependent effect. This is where is gets complicated, because neither the H3 nor N2 protein alone can recapitulate the original X31 phenotype. Infection of DAF-knockout mice with the H3-only virus results in lower C3a concentrations, and reduced infiltration of innate immune cells 3 and 6 dpi (monocytes are not significant 3 dpi, but this is likely due to a single outlier). In addition, T-cell numbers are reduced 6 dpi. However, the effects on weight loss are greatly reduced and there is no effect on survival. Infection of DAF-knockout mice with the N2-only virus also reduced C3a concentration, and neutrophils and monocyte infiltration 3 dpi. No effect on innate immune cell infiltration 6 dpi was observed. Unlike H3-only virus infection, N2-only infected DAF-knockout mice lost less weight and all the animals survived. This raises several questions: what is the role of C3a in all this, iis the reduction in C3a concentration related to the phenotype, and is the C3a concentration dependent on the virus used?

Finally, they demonstrate that NA can cleave sialic acids from DAF resulting in lower C5b-C9 deposition on cells. While this is an interesting observation, this effect does not seem to correlate with the in vivo data in the same manuscript. The NA of PR8 cleaves DAF very well, but it did not have a phenotype in the DAF-knockout animals.

To summarize, this manuscript reports on two very interesting observations, i.e. role of DAF in influenza disease, and de-sialylation of DAF by NA. However, the latter does not appear to explain the first observation, resulting in a lack of understanding of either observation. Inclusion of one or more of the following experiments may shed light on the connection between these two observations or produce additional insight into the mechanism behind the DAF-dependent effects on influenza disease.

1) Determine the C3a concentration over time in WT and DAF-knockout mice after infection with PR and X31.

2) Determine histology score and immune cell infiltration after PR8 infection in WT and DAF-knockout mice.

3) Test additional “7+1” viruses where you compared additional NA proteins, that either cleave or do not cleave sialic acids from DAF in order to link that observation to an in vivo phenotype.

4) The depiction of C3 in the model figure is not fully supported by the data. How is HA expression linked to complement activation? Same with sialic acid cleavage of DAF by NA. Because different doses of virus were used, it is difficult to compare C3a levels between PR8, X31, and the H3 and N2 only viruses. Without this information, you cannot conclude that sialic acid cleavage of DAF affects activity in vivo and thus C3a concentration.

Minor edits:

1) Normalizing the flow data to WT may exacerbate small differences. It is better to present the actual cell numbers.

2) Is A/England/159/2009 also a pdmH1N1 virus and if so, why is the phenotype different from A/Cal/7/2009?

3) You mention for Fig 1D that the survival is 25% and it is more like 30%

4) Consider legend location for Fig 3 between A-B and D-F (horizontal).

5) Fig legend 2S does not appear to have a F and G section.

**Part II – Major Issues: Key Experiments Required for Acceptance**

Reviewer #1: Fig 2c – Can the authors describe what the end point in their survival experiment was? In the methods is appears a weight loss >25% was grounds for euthanasia. If this is the case, it is unclear how the WT mice which lost less weight compared to the Daf-/- mice have worse survival rates. Can the individual weight loss curves for each mouse be presented instead.

Fig 3b-c – To exclude DAF depletion does not impact viral clearance the authors should extend the time course and show that both cohorts clear the virus at the same rate (ie a later time point showing viral loads is required).

Fig 4a – It is unclear as to why the authors dropped the viral dose in this experiment to 500PFU PR8-HK4,6. Moreover, the control of infection of the Daf-/- with 500 PFU PR8-HK4,6 is missing and needs to be added to this dataset to show at this dose Daf deficiency still improves disease outcomes.

Fig 4b – Assessment of C3a levels in the BAL at a single time point is insufficient to conclude that IAV triggers lower levels of C3a in DAF deficient mice – perhaps levels could have peaked earlier in this cohort. Assessment of C3a levels in the BAL at d0, 3, 6, 10 p.i. would be more convincing evidence for lower complement activation in Daf mice post flu.

Fig 4d-j – Assessing immune cell infiltration into the BAL alone following IAV infection of Daf-/- vs WT mice is an extremely narrow analysis. It is important that the immune cells infiltrating the lung tissue are also quantified. Moreover, it not clear why the authors chose to normalise the data generated using BAL samples to the WT sample, especially when the variability in the WT group ranged by +/-50%. Absolute numbers/ml of BAL would be more informative.

Fig 4k – the assessment of a single cytokine (IFNg) at a single time point (d6) is insufficient to conclude that there is less inflammatory cytokine production in the Daf-/- mice post influenza. Assessment of a panel of proinflammatory cytokines (including IL-6 which is mentioned but not measured) and chemokines using a bead array type approach will help significantly strengthen this conclusion.

Fig 6 a- the control of the DAf-/- alone is missing from this analysis, and a broader time period of assessment for C3 analysis in the BALF is required.

Line 324 – the statement “ that HA-DAF interaction modulates the adaptive immune response “ needs to be tempered or further experimental analysis needs to be performed to support this. This statement is based on assessment of bulk CD4 and CD8 in BALF at a single time point. Either track antigen specific CD4 and CD8 responses over the course of the infection or temper the statement.

Reviewer #2: 1) Determine the C3a concentration over time in WT and DAF-knockout mice after infection with PR and X31.

2) Determine histology score and immune cell infiltration after PR8 infection in WT and DAF-knockout mice.

3) Test additional “7+1” viruses where you compared additional NA proteins, that either cleave or do not cleave sialic acids from DAF in order to link that observation to an in vivo phenotype.

4) The depiction of C3 in the model figure is not fully supported by the data. How is HA expression linked to complement activation? Same with sialic acid cleavage of DAF by NA. Because different doses of virus were used, it is difficult to compare C3a levels between PR8, X31, and the H3 and N2 only viruses. Without this information, you cannot conclude that sialic acid cleavage of DAF affects activity in vivo and thus C3a concentration.

**Part III – Minor Issues: Editorial and Data Presentation Modifications**

Reviewer #1: Fig 4d- scale has a typo

Table S1 – Some representative histological profiles showing examples of disease pathology would be helpful

Fig 5-7 – the results generated with the hybrid PR8-HK6 or PR8-HK4 are confounded by the increased virulence of these strains compared to PR8-HK6,4 which sees every set of experiments performed with different doses of viruses. The authors should acknowledge this caveat. Moreover, this section is difficult to understand and I recommend the authors try simplifying the message

Reviewer #2: Minor edits:

1) Normalizing the flow data to WT may exacerbate small differences. It is better to present the actual cell numbers.

2) Is A/England/159/2009 also a pdmH1N1 virus and if so, why is the phenotype different from A/Cal/7/2009?

3) You mention for Fig 1D that the survival is 25% and it is more like 30%

4) Consider legend location for Fig 3 between A-B and D-F (horizontal).

5) Fig legend 2S does not appear to have a F and G section.

PLOS authors have the option to publish the peer review history of their article (what does this mean?). If published, this will include your full peer review and any attached files.

Reviewer #1: No

Reviewer #2: No
---

## [Decision Letter · Decision Letter 1]

26 May 2021

Dear Dr. Amorim,

We are pleased to inform you that your manuscript 'Complement Decay-Accelerating Factor is a modulator of influenza A virus lung immunopathology' has been provisionally accepted for publication in PLOS Pathogens.

Best regards,

Jacob S. Yount

Guest Editor

PLOS Pathogens

Carolina Lopez

Section Editor

PLOS Pathogens

Kasturi Haldar

Editor-in-Chief

PLOS Pathogens

orcid.org/0000-0001-5065-158X

Michael Malim

Editor-in-Chief

PLOS Pathogens

orcid.org/0000-0002-7699-2064

Reviewer Comments (if any, and for reference):

Reviewer's Responses to Questions

**Part I - Summary**

Reviewer #1: The authors have adequately addressed my concerns.

**Part II – Major Issues: Key Experiments Required for Acceptance**

Reviewer #1: The authors have adequately addressed my concerns.

**Part III – Minor Issues: Editorial and Data Presentation Modifications**

Reviewer #1: (No Response)

PLOS authors have the option to publish the peer review history of their article (what does this mean?). If published, this will include your full peer review and any attached files.

Reviewer #1: No

---

## [Editor Report · Acceptance letter]

7 Jun 2021

Dear Dr. Amorim,

We are delighted to inform you that your manuscript, "Complement Decay-Accelerating Factor is a modulator of influenza A virus lung immunopathology," has been formally accepted for publication in PLOS Pathogens.

Best regards,

Kasturi Haldar

Editor-in-Chief

PLOS Pathogens

orcid.org/0000-0001-5065-158X

Michael Malim

Editor-in-Chief

PLOS Pathogens

orcid.org/0000-0002-7699-2064